REGISTERED REPORT PROTOCOL

# Registered Report Protocol: Survey on attitudes and experiences regarding preregistration in psychological research

**Lisa Spitzer** *, Stefanie Mueller

Leibniz Institute for Psychology, Trier, Germany

* ls@leibniz-psychology.org

## Abstract

### Background

Preregistration, the open science practice of specifying and registering details of a planned study prior to knowing the data, increases the transparency and reproducibility of research. Large-scale replication attempts for psychological results yielded shockingly low success rates and contributed to an increasing demand for open science practices among psychologists. However, preregistering one's studies is still not the norm in the field. Here, we propose a study to explore possible reasons for this discrepancy.

### Methods

In a mixed-methods approach, an online survey will be conducted, assessing attitudes, motivations, and perceived obstacles with respect to preregistration. Participants will be psychological researchers that will be recruited by scanning research articles on Web of Science, PubMed, PSYNDEX, and PsycInfo, and preregistrations on OSF Registries (targeted sample size: $N = 296$). Based on the theory of planned behavior, we predict that positive attitudes (moderated by the perceived importance of preregistration) as well as a favorable subjective norm and higher perceived behavioral control positively influence researchers' intention to preregister (hypothesis 1). Furthermore, we expect an influence of research experience on attitudes and perceived motivations and obstacles regarding preregistration (hypothesis 2). We will analyze these hypotheses with multiple regression models, and will include preregistration experience as control variable.

## Introduction

Ever since Ioannidis [1] argued that most published research findings are false and a multitude of attempts failed to replicate previously significant effects (e.g., [2]), the reliability of published research findings has been a subject of discussion across many scientific disciplines. Summarizing these concerns, the term *replication crisis* [3] arose, where *replicability* refers to the attempt to repeat an experiment to re-test the original effect [4]. When *Nature* [5] conducted a survey of more than 1500 researchers of multiple disciplines, 70% of researchers reported that

**Data Availability Statement:** We have uploaded the pilot study data and the code for analyzing it on PsychArchives (https://www.psycharchives.org/), a platform for sharing psychological data sets and other digital research materials. The data of the pilot study (including meta-data about variables and values) is accessible via the DOI: http://dx.doi.org/10.23668/psycharchives.4798 The analysis scripts for the pilot study are accessible via the DOI: http://dx.doi.org/10.23668/psycharchives.4799 Additionally, we will also share all data of the main study on PsychArchives after study completion.

**Funding:** The authors received no specific funding for this work.

**Competing interests:** The authors have declared that no competing interests exist.

they had failed to replicate studies by others, and more than 50% had failed to replicate their own studies. Overall, 90% of researchers indicated their belief in a slight or even significant crisis [5]. In psychology, multiple large-scale research projects attempted to replicate significant effects published in top tier journals. Strikingly many attempts failed as shown by replication rates between 36 [2] and 77% [6]. Among successfully replicated effects, the effect sizes were considerably smaller than originally reported.

It has been reasoned that false positives, i.e., effects that are significant in studies but do not exist in reality, contribute to low replicability [7]. The high rate of false positive research results has largely been attributed to "questionable research practices" (e.g., see [8,9]), a collective term for any "exploitation of the gray area of acceptable practice . . . (which can) increase the likelihood of finding evidence in support of a hypothesis" ([9] p. 524). Examples for these practices are the failure to control for biases, selective reporting of significant results, $p$-hacking, or revising the hypotheses to match the results, also known as HARKing (see [8,10–14 for details).

## Preregistration–on the rise?

The preregistration of studies has been proposed to counter these questionable research practices (e.g., see [15], and see [10] for an overview of other open science techniques). A preregistration is a research plan that is time-stamped, created before the data has been collected or examined, and most often submitted to a public registry, thus making planned study details available to others (possibly after an embargo period) [15,16]. If the research plan changes afterwards, either a new version needs to be added or the deviations will be apparent when comparing the preregistration to the final manuscript. Thus, preregistration aims for a transparent presentation of what was planned at a certain time point and what changes may have been made to a study until its publication. Evidence from other scientific disciplines indicates that preregistration indeed works, i.e., it increases the transparency of the research process, and reduces questionable research practices and the rate of false positive findings (e.g., see [17,18]).

However, while preregistration is already well-established in other scientific disciplines and is mandated, for example, in medicine [19], it has been frequently demanded as a means to counter questionable research practices but is still not widely practiced in psychology.

On the one hand, in response to the replication crisis, many psychologists have committed themselves to the advancement and promotion of open science techniques such as preregistration (e.g., see [10,13,15,20–22]). For example, Nosek et al. [20] describe preregistration as "hard, and worthwhile", while Wagenmakers and Dutilh [22] posit "seven selfish reasons for preregistration". Indeed, the number of preregistrations in psychology is increasing. For instance, the number of preregistrations on the Open Science Framework (OSF), a platform for sharing research materials, has been approximately doubling every year between 2012 and 2017 [21], and in a survey which was conducted in 2018, 44% of the sampled psychological researchers indicated having preregistered a hypothesis or analysis until 2017 [23].

Yet, looking at the fraction of published studies that were actually preregistered paints a different picture. In their recent study, Hardwicke et al. [24] found that only 3% of 188 examined articles from 2014 to 2017 included a preregistration statement, which contradicts the more positive outlook by [21] and [23]. Stürmer et al. [25] also found that when early career researchers were asked about questionable research practices and open science, they deemed many open science practices necessary, yet toward preregistration they expressed more reluctance: Only about half of the participants found that preregistration was fairly necessary or very necessary, and even less indicated that they planned to consider preregistering their studies in the near future.

A number of reservations are mentioned frequently when discussing preregistration in psychology, including the fear that it leaves no flexibility during study administration and eliminates the possibility to conduct exploratory analyses (as presented by e.g., [15,16]). Some people are concerned that this might stifle discovery [26]. Others worry that someone might take their preregistered and thus, publicly available, study idea and publish it before them (so-called scooping, see [27]). Additionally, the time costs and effort are often seen as obstacles regarding preregistration (e.g., see [16]). Besides these worries, some authors also express an overall critique regarding the concept of preregistration. Szollosi et al. [28–30] argue that preregistration is redundant when good theories are tested, and does not itself improve theories. Others argue that preregistration cannot prevent some questionable research practices [31], or might not fit well with all types of research (see [16]). Moreover, some studies found problems with the current implementation of preregistration such as poor disclosure of deviations from preregistered plans in finished manuscripts [32–34]. Although most of the listed arguments against preregistration are counter-argued by supporters of preregistration (e.g., that exploratory analyses are still possible [15,16,27]) and findings from other scientific disciplines underline its benefits (e.g., see [17,18]), these reservations persist and some researchers remain skeptical.

## Aim of this survey

While previous surveys inquired about preregistration in the more general context of open science [23,25,35,36], to our knowledge no comprehensive study focusing on preregistration has been conducted. We aim to close this gap by exploring thoughts, motivations, and perceived obstacles of psychological researchers toward preregistration and how these are influenced by the time someone has worked in research or actual experience with preregistration, that is, whether someone preregistered a study in the past. For instance, we want to explore the data to find out whether the low rate of preregistrations is caused by fear of the unknown or based on negative experiences (and which ones), as well as whether the increase of preregistrations is driven by a few active supporters while others reject or are indifferent toward it. Thus, we aim to shed light on the outlined discrepancy of public support for preregistration on the one hand, and a low fraction of preregistrations on the other, while also identifying possible roadblocks for preregistration in psychology. Mixed-methods will be used, including both qualitative and quantitative approaches.

Additionally, we want to investigate two specific research questions: First, we want to examine which factors facilitate or prevent preregistration (*research question 1*). The theory of planned behavior [37,38] is a prolific, influential (e.g., see [39]) and well-researched (e.g., see [40–45]) psychological theory that aims to predict social behavior and has been applied across various contexts (e.g., health). According to this theory, the intention to perform a behavior can be seen as a direct antecedent of the actual behavior. In this framework, the intention to preregister predicts preregistration, and how the intention is formed may be informative for effectively promoting this behavior. To our knowledge, this has not yet been studied in the context of preregistration or open science. As described by Ajzen and colleagues [37,38], three aspects influence intentions which are defined as follows: 1) Attitudes toward the behavior which result from the ratio of perceived advantages to disadvantages of performing the behavior, 2) the subjective norm which represents the perceived social pressure to perform or not perform the behavior, and 3) the perceived behavioral control which focuses on the question if the subject has the resources and skills to perform the behavior or not (also see [40–45] for meta-analytical support of this model, and [46] for an overview). We will measure attitudes toward preregistration as well as subjective norms and perceived behavioral control through

items in an online questionnaire, and investigate how they influence researchers' intention to preregister their studies in the future. Based on the model's postulations, we expect that more favorable attitudes and subjective norms as well as higher perceived behavioral control positively influence the intention to use preregistration. As the relative importance of attitudes, subjective norms and perceived behavioral control differs in dependence of considered behaviors, situations and populations [37,38], we want to test which of these is the strongest predictor for the intention to use preregistration. We will also include the perceived importance of preregistration as moderator for the strength of the influence of attitudes on intention, and the preregistration experience as a control variable. Such an extension of the model will compensate for potential non-attitudes (e.g., see [47]) and a potential sampling bias toward researchers that have already preregistered, and is explicitly allowed by the theory of planned behavior [37,46].

Regarding the intention formation, we have the following hypotheses:

1. The theory of planned behavior [37,38] can be applied to the context of preregistration to significantly predict researchers' intention to preregister their studies in the near future, using a moderated multiple regression model. We predict that:

    1.1. More beneficial attitudes are a positive predictor for the intention to preregister.

    1.2. The perceived importance of preregistration moderates the effect of attitudes on intention positively.

    1.3. The perceived importance of preregistration is a positive predictor for the intention to preregister.

    1.4. Beneficial subjective norms are a positive predictor for the intention to preregister.

    1.5. Higher perceived behavioral control is also expected to be a positive predictor.

    1.6. These predictors combined can significantly predict researchers' intention to preregister.

Second, we want to examine whether research experience predicts attitudes and the perceived intensity of motivations and obstacles (*research question 2*). Research experience will be operationalized as the number of years someone indicates they have worked in psychological research. Early career researchers are oftentimes seen as the driving force of the open science movement (e.g., see [48]). We want to investigate if the research experience indeed has an influence on researchers' responses about preregistration (a similar effect was reported by Abele-Brehm et al. in a comparison of academic groups regarding hopes and fears toward open science [35]).

Regarding this second research question, the following hypotheses will be tested:

2. We predict that research experience, that is, the amount of time someone has already worked in psychological research, has an influence on attitudes, motivation, and perceived obstacles regarding preregistration. Specifically, we will conduct three multiple regressions (including preregistration experience as control variable), and we posit the following non-directional hypotheses:

    2.1. Research experience is a predictor for attitudes regarding preregistration.

    2.2. Research experience is a predictor for the strength of motivation to preregister.

    2.3. Research experience is a predictor for how strongly obstacles to preregister are perceived.

The present survey is aimed to sample the general population of psychological researchers by recruiting participants whose articles appear on Web of Science, PubMed, PSYNDEX, and

PsycInfo, as well as the subgroup of researchers who have preregistered before and who will be identified through their preregistration on OSF Registries.

## Methods

### Sampling procedure

**Power analyses.** Data from psychological researchers at different career stages will be collected. The optimal sample size reported below has been determined by using G*Power [49,50] in combination with a thorough review of the existing literature as described in the following paragraphs. Each of the power analyses described below was specified to achieve a statistical power of 95% at a given significance threshold of 5% ($\alpha = \beta = .05$). All power analyses are displayed in Fig 1 and are also included in the supporting information (see S1 Text).

To test which factors influence the intention to preregister (scale of three items, see *hypothesis 1*), a moderated multiple regression model will be computed based on the rationale of the theory of planned behavior, which includes six predictors: Attitudes (scale of 24 items), perceived importance of preregistration (one item in our questionnaire), attitudes x importance, subjective norm (scale of eight items), perceived behavioral control (scale of five items) [37,38,46], and preregistration experience (one item). The theory of planned behavior has been examined using meta-analytical approaches in various contexts (e.g., health behavior). The percentage of variance of intention that was explained by attitudes, subjective norm and perceived behavioral control combined, ranged between 30.4% $< R^2 <$ 44.3% [40,42–45]. We chose the lowest reported effect size ($R^2 = 30.4\%$) as minimal effect size of interest. The power analysis for the overall regression model yielded an optimal sample size of $N = 55$ to be able to

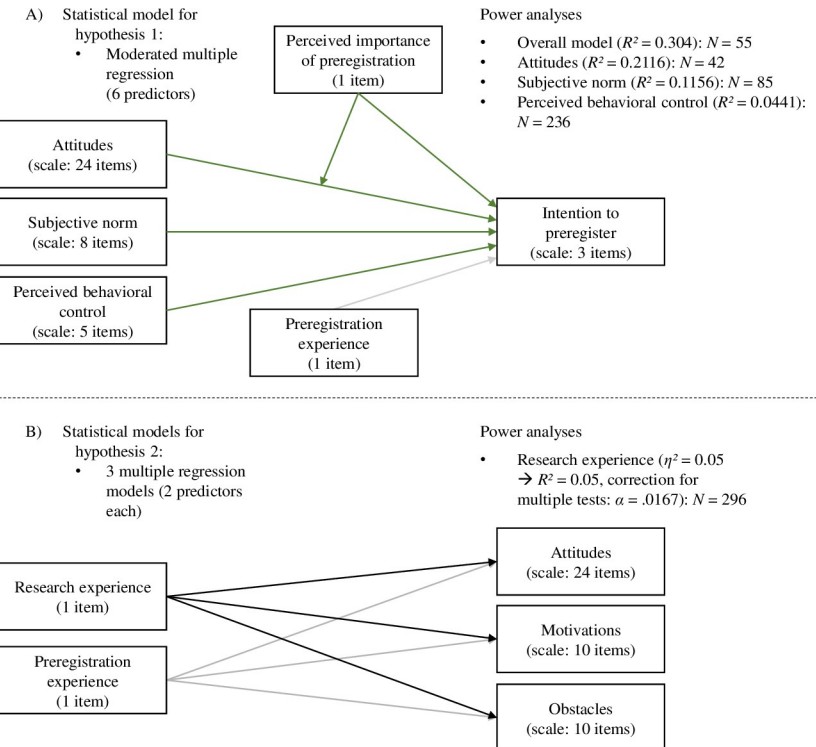

**Fig 1. Overview of the power analyses.** The detailed statistical models and corresponding power analyses are displayed for A) the moderated multiple regression that will be computed for testing hypothesis 1 and B) the multiple regressions that will be used to test hypothesis 2.

detect the determined effect size given the effect exists (see *hypothesis 1.6*). Additional power analyses were conducted to compute the optimal sample size to test the individual predictors. As comparable effect sizes, $R^2$ based on the averaged correlations of individual variables were searched for in the aforementioned meta-analyses, and the smallest ones were chosen for each power analysis. This resulted in an optimal sample size of $N = 42$ for testing attitudes (see *hypothesis 1.1*), $N = 85$ for testing subjective norms (see *hypothesis 1.4*), and $N = 236$ for testing perceived behavioral control (see *hypothesis 1.5*) as predictors for intention (with $\alpha = \beta = .05$). Since perceived importance and its interaction with attitudes, as well as preregistration experience are not originally included in the model but added by us, no comparable effect sizes are available and thus, no power analyses have been conducted for these variables.

To investigate whether research experience has an effect on attitudes (see *hypothesis 2.1*), and the perceived intensity of motivations (see *hypothesis 2.2*) and obstacles (see *hypothesis 2.3*) regarding preregistration, three regression models will be computed, again including preregistration experience as control variable. As comparable effect size, the smallest reported effect size for group differences by Abele-Brehm et al. [35] was used, and the corresponding $F$ value was used to calculate $R^2$ for the power analyses. For these regression models, the alpha level will be corrected using the Bonferroni-Holm method to account for multiple testing. Thus, the power analyses were calculated for the smallest alpha of 1.67% (5% divided by 3). These power analyses suggested an optimal sample size of $N = 296$ to test the influence of research experience on attitudes (see *hypothesis 2.1*), perceived intensity of motivations (see *hypothesis 2.2*), and perceived intensity of obstacles (see *hypothesis 2.3*) regarding preregistration.

As the optimal sample size for the regression models to test hypothesis 2 ($N = 296$) is the highest computed necessary *N*, it will constitute the targeted sample size.

**Data collection and stopping rule.** To ensure that our sample contains researchers with varying research experience, a quota sampling will be used to evenly collect data from researchers with different degrees, that is, bachelor's degree, master's degree, doctoral degree, or habilitation and/or full professorship. Specifically, a quota sampling will be used with a 25% quota for each subgroup. Yet, as it might be the case that some quotas might not be filled, quotas are defined for a slightly bigger sample than the necessary $N = 296$ to compensate for this eventuality. In particular, $N = 400$ will be the basis for our quotas which will therefore be $n = 100$ for each group. This enables us to reach the a priori computed power even if not all quotas can be filled, while still ensuring that no group is overrepresented.

To recruit this sample, a method similar to the one by Field et al. [51] will be used. The term "psychology" will be searched for on specified databases, and resulting hits will be sorted from newest to oldest. These documents will be scanned for authors whose email addresses can be found via institutional or personal websites linked to the respective work, or by searching for the author via Google, Google Scholar, and ResearchGate. Duplicated email addresses that may be sampled from different databases will be excluded. Identified persons will be invited to participate in the survey via email.

For this search, the databases Web of Science (https://apps.webofknowledge.com/), PubMed (https://pubmed.ncbi.nlm.nih.gov/), PSYNDEX (https://www.psyndex.de/), and PsycInfo (https://www.apa.org/pubs/databases/psycinfo) will be used to recruit a representative sample of psychological researchers.

Based on the results by Hardwicke et al. [24], we anticipate that only a small proportion of the research articles that will be found on the general databases will include a preregistration statement, which might lead to a small number of participants with any preregistration experience. Thus, we decided to send 10% of invitations to authors of preregistrations in order to ensure that our survey will also include a sufficient number of participants who have preregistered before. These participants will be identified via the preregistration platform OSF

Registries (https://osf.io/registries). As this may introduce a sampling bias towards researchers who lean more positive towards preregistration, preregistration experience will be controlled for in the statistical analyses aimed to draw inferences about the general population of psychological researchers.

A customized link will be distributed among participants of each database so that the recruitment source can be inferred. The database participants were recruited from will be considered in the analyses as described in the section *descriptive reports*.

Response rates were rather low in many studies that relied on researchers as sample (e.g., [9,25,35,36,51–53]). Based on these studies and based on the insights from our pilot study (see section *pilot study*), we anticipate a response rate of around 10%. To compensate for this, we will invite $N$ = 2960 persons ($n$ = 666 via Web of Science (22.5%), $n$ = 666 via PubMed (22.5%), $n$ = 666 via PSYNDEX (22.5%), $n$ = 666 via PsycInfo (22.5%), and $n$ = 296 via OSF Registries (10%)) to reach our target sample size of $N$ = 296 (10% of 2960). If this does not result in a sufficient sample size, an additional invitation wave will be conducted. In this second wave, participants will subsequently be invited for quotas that have not yet been filled. For each open quota, the response rate of the first invitation wave will be used to calculate how many more participants need to be invited to fill the quota. This procedure is further described in Fig 2, and in more detail in the supporting information (see S2 Text). Individuals of the first wave will have approximately one month overall to participate in the survey, individuals of the second wave will have two weeks to participate. Both groups will receive a reminder email one week after invitation. Additionally, the survey will be advertised on social media (Facebook, Twitter), and via student and researcher specific mailing lists, which will be seen as additional sources of participants.

If the optimal sample size is achieved before the end of the set time frame (approximately one month), the survey will still be accessible to participants that were already invited, but recruitment will be discontinued. Data collection will be stopped after this time frame even if the optimal $N$ cannot be reached.

The collection of email addresses has started on January 4th, 2021. Emails will be sent once our submission receives an in-principle-acceptance and 2960 email addresses have been collected using the procedure described above.

Participants who completed the survey will have the possibility to participate in a lottery for 40 gift cards worth 50 € each. The survey was approved by the ethics committee of Trier University, Germany.

**Exclusion and missing data.** All participants that indicate that their research or studies do not fall within the scope of psychology or do not have at least a bachelor's degree in psychology, and thus, cannot be assigned to the respective quota, will be screened out at the beginning of the survey. Thus, they will be directed to an exit page rather than to the main body of the survey, will not be counted into the quotas, and will not be considered for data analyses as the survey targets a sample from research-oriented psychology.

At the end of the survey, participants will be asked whether they responded faithfully. Only data of participants that indicated having at least a bachelor's degree in psychology, indicated faithful participation, and completed all pages will be counted for quota fulfillment. By screening the open text inputs, it will additionally be checked if participants included any inappropriate responses like advertising or offensive comments. Those participants will be excluded from analyses. Data from all participants who indicated faithful participation, who were not excluded based on inappropriate answers, and completed all pages of the survey will be analyzed when testing the planned hypotheses (see section *hypotheses tests*). Additionally, responses from participants who were not screened out or excluded based on inappropriate responses, and completed the survey only partially, will be used for calculating descriptive statistics where applicable (see section *descriptive reports*). A sensitivity analysis will be performed

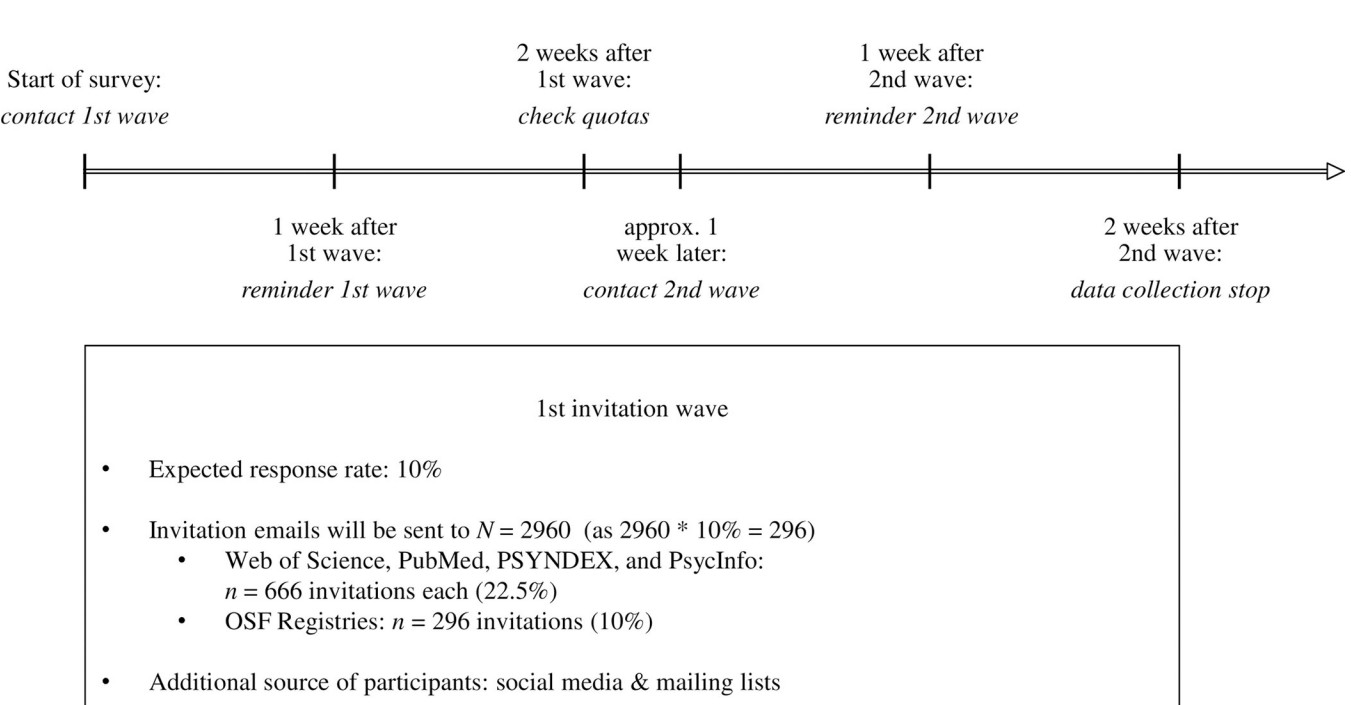

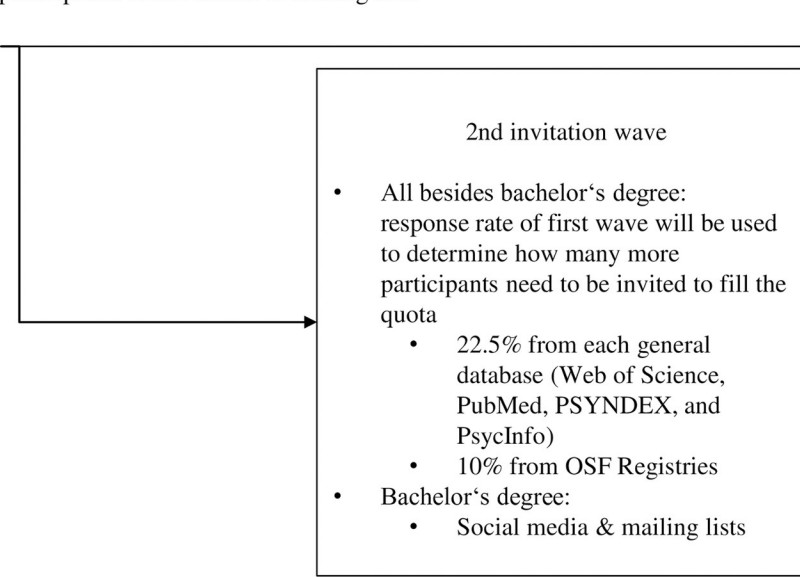

**Fig 2. Survey timeline and corresponding actions.** The arrow illustrates the timeline while the boxes contain details about the sampling procedure.

to examine whether and to what extent the exclusion of incomplete datasets for the hypotheses tests affects the results.

## Material

The online survey was created with the software SoSci Survey (version 3.2.29) [54] and will be supplied via www.soscisurvey.de. It will be presented in English. Items of the survey can be categorized in nine categories: 1) Sociodemographic questions, 2) items concerning the general usage of preregistration, 3) an *attitude* scale consisting of 24 items indicating the overall attitudes of participants regarding the concept of preregistration, 4) a *subjective norm* scale of eight items representing perceived social norms and pressure, 5) a *perceived behavioral control* scale of five items inquiring about researchers' perceived control over the potential preregistration of their studies, 6) an *intention* scale of three items inquiring about researchers' intention to use preregistration in the future, 7) items about motivations (a *motivation* scale including ten items measuring how strongly participants agree with potential motivations to preregister, plus additional more open items), 8) items about perceived obstacles (an *obstacle* scale featuring ten items measuring how strongly participants agree with potential obstacles to preregister, plus additional more open items), and 9) various open questions inquiring about suggestions for improving preregistration. Some items were adapted from other surveys that focused on related topics [25,35] and are complemented with additional items generated in reference to other theoretical and empirical works on preregistration, open science [31,51,53,55], and the theory of planned behavior [37,38,46,56]. These sources were selected based on their theoretical relevance to the topic at hand (i.e., open science practices and theory of planned behavior). The identified items were either adapted (i.e., wording was changed slightly to make them easier to understand or make them fit the present topic, e.g., "I have more trust in research findings when the study has been preregistered." instead of ". . . when the respective data are published." [35]) or new items were generated using the concepts presented in the literature based on face validity. The items of the present survey as well as the original items including references are available in the supporting information (see S1 Table). For most of the original items, no validity measures are given in the literature. Only for the theory of planned behavior, predictive validity is considered in more detail [46]. However, since none of the survey items were used in their original form but were adapted or newly created based on the given concepts, no validity measures can be provided. Instead, only face validity is assumed. This validity evaluation is based on Flake and Fried [57].

Different item formats will be included in the survey. All scale items (attitudes, subjective norm, perceived behavioral control, intention, motivations, and obstacles) will be answered with a seven-point labeled answer scale as recommended by Ajzen [37] (1 = "Strongly disagree", 2 = "Disagree", 3 = "Slightly disagree", 4 = "Neither agree nor disagree", 5 = "Slightly agree", 6 = "Agree", 7 = "Strongly agree"). Scales will be recoded from "1 to 7" to "-3 to +3" for data analyses yielding a middle category which has absolute meaning. For the statistical analyses, the mean scores of the scales will be used to measure how participants perceive 1) preregistration (attitude scale), 2) subjective norms regarding preregistration (subjective norm scale), 3) their own control about using preregistration or not (behavioral control scale), 4) their intention to use preregistration in the future (intention scale), and 5) their motivations (motivation scale) and 6) obstacles to preregister (obstacle scale). For each participant, the mean for each scale will be calculated and used as the score.

Other items will include a single or multiple choice response format, or the option for open text input. Whenever applicable, options are displayed in randomized order to eliminate potential sequence effects.

## Procedure

Participants will receive the link to the survey via personal email or social media call. After the welcoming page, participation information will be displayed and informed consent needs to be provided to proceed. Furthermore, a captcha (arithmetic task) needs to be completed as a safeguard against bot responses. Then, participants will complete the main body of the survey which successively focuses on the different item categories (sociodemographic questions, general usage of preregistration, attitudes, subjective norms, perceived behavioral control, intention, motivations, perceived obstacles, and suggestions for improvement). Before any items related to preregistration are shown, a definition of preregistration is presented and correct understanding is checked, to ensure that all participants answer the items with the same concept in mind. At the end of the survey, participants will have the option to participate in the lottery and to sign up for receiving a preprint of the survey results by entering their email address, which will be saved separately from the other data. Lastly, a debriefing will follow that states the aim of the survey and the research questions that are investigated. Overall, completing the survey will take about 20–25 minutes. Screen recordings of the survey's procedure are available in the supporting information (see S1 and S2 Videos).

## Data analysis

This Registered Report Protocol was written in *R Markdown* (version 2.7) [58,59]. R (version 4.0.5) [60] and the R-packages *afex* (version 0.27–2) [61], *Amelia* (version 1.7.6) [62], *beeswarm* (version 0.2.3) [63], *biotools* (version 3.1) [64], *car* (version 3.0–8) [65,66], *carData* (version 3.0–4) [66], *corrplot2017* (version 0.84) [67], *dplyr* (version 1.0.0) [68], *emmeans* (version 1.4.7) [69], *forcats* (version 0.5.0) [70], *ggplot2* (version 3.3.2) [71], *gplots* (version 3.0.3) [72], *heplots* (version 1.3–5) [73,74], *lattice* (version 0.20–41) [75], *lm.beta* (version 1.5–1) [76], *lme4* (version 1.1–23) [77], *MASS* (version 7.3–53.1) [78], *Matrix* (version 1.3–2) [79], *MBESS* (version 4.7.0) [80], *mlbench* (version 2.1–1) [81], *MVN* (version 5.8) [82], *olsrr* (version 0.5.3) [83], *papaja* (version 0.1.0.9942) [84], *popbio* (version 2.7) [85], *psych* (version 1.9.12.31) [86], *purrr* (version 0.3.4) [87], *Rcpp* (version 1.0.4.6) [88,89], *readr* (version 1.3.1) [90], *reshape2* (version 1.4.4) [91], *rpanel* (version 1.1–4) [92,93], *sp* (version 1.4–2) [94], *SpatialEpi* (version 1.2.3) [95], *stringr* (version 1.4.0) [96], *tibble* (version 3.0.1) [97], *tidyr* (version 1.1.0) [98], *tidyverse* (version 1.3.0) [99], *tkrplot* (version 0.0–24) [100], and *writexl* (version 1.3) [101] will be used for the analyses. Once the study has been conducted, StatTag (version 6.0.1) [102] will be used to insert the results calculated in R into the manuscript. Scripts for preprocessing, assumption testing and all analyses will be available in the supporting information (see S1–S3 Scripts). Additionally, all data (including meta-data about variables and values) will be publicly accessible after study completion in PsychArchives (https://www.psycharchives.org/), a platform for sharing psychological data sets and other digital research materials.

**Preprocessing.** Data will be processed in the following way: Responses from all scale items will be recoded as described in the section *material*, and polarity of negatively poled items will be reversed. Additionally, multiple choice questions will be recoded (originally: 1 = "not checked" and 2 = "checked"; new: 0 = "not checked" and 1 = "checked"), and single choice items will be transformed into factors. Comments will be inspected. Furthermore, data will be screened for any inappropriate responses (i.e., advertising, offenses) which would lead to an overall exclusion.

As a quality check, the attitude, subjective norm, perceived behavioral control, intention, motivation, and obstacle items will be inspected with respect to floor or ceiling effects which will then be excluded from the analyses (specifically, items for which $\geq$ 90% of participants indicated the lowest or highest category). Additionally, Cronbach's alpha will be calculated as

an indicator for reliability for each scale, and items for which Cronbach's alpha of the scale is at least .10 higher if they are dropped will be excluded. The remaining items will be used to calculate mean scores for the attitude, subjective norm, perceived behavioral control, intention, motivation, and obstacle scale respectively for each participant. For this quality check, only complete datasets will be considered. All preprocessing scripts are available in the supporting information (see S1 Script).

**Descriptive reports.** To describe the general behavior of participants when filling out the survey, we will calculate the response rate, the number of screened out participants, the number of complete and incomplete survey participations, the completion time (*M*, *SD*, *range*), and the database from which participants were recruited.

We will further describe the composition of the obtained sample by providing frequency tables or plots and descriptive statistics (*M*, *SD*, *range*) for selected sociodemographic information (e.g., gender, age, country of residence, academic degree, main research topic, number of years worked in research) separately for participants who preregistered in the past vs. participants without preregistration experience. Means, standard deviations, and ranges for perceived importance of preregistration and intention to preregister in the near future will be calculated. We will furthermore compare these between researchers with different academic degrees. If the respective group means do not differ as determined by a one-factorial ANOVA, we will report the overall mean, standard deviation and range across groups.

To gain insight into how common preregistration is in the general population of psychological researchers, we will report the mean proportion of participants who used preregistration in the past and how many preregistrations they created on average, across the sample obtained from the article databases, that is, excluding participants recruited from OSF Registries.

Moreover, also including the sample recruited from OSF Registries, participants' responses to the following questions will be listed as frequency tables or in the text: 1) Where did they learn about preregistration, 2) what motivated them to create their first preregistration (only addressing participants with preregistration experience), 3) what problems did researchers experience who already used preregistration, and in turn, what worries do researchers report who have not preregistered yet (and do these match?), 4) how often do participants read other researchers' preregistrations, and 5) by whom is the researchers' decision to preregister influenced the most, which is compared between researchers with different academic degrees. Wherever applicable, the results will be compared between participants that have preregistered before or not.

Open text inputs of both the open "other" options of selection items and the open text input items (i.e., what are benefits and drawbacks of preregistration, what might be positive and negative consequences of mandatory preregistration, what are reasons against preregistration, how and why did participants' motivation to preregister change, and what are suggestions for increasing the motivation, decreasing obstacles, and for other improvements regarding preregistration) will be analyzed to identify common themes. The following mixed-methods approach will be used: Two to four coders will qualitatively identify themes mentioned by the participants and subsequently categorize all responses accordingly in order to receive a frequency measure that will be analyzed quantitatively. Specifically, open text inputs of each item will be shuffled and the first 10% of these shuffled responses will be used to establish initial categories of themes. Coders will read the responses and add each theme as a column in a coding sheet. It will be coded whether the respective theme appears in the other responses (= 1) or not (= 0). If a coder encounters new relevant themes, they will be added and coded later. Nonsense responses will be excluded during coding. Ambiguities will be discussed and solved in pairs. If no solution can be found, a third coder will be consulted. After all responses have been coded, the sums for each column (= theme) will be calculated to obtain the frequency of how often a

theme was mentioned over all responses. Results from the open text input items will be displayed in frequency tables, and the results of the analysis of "other" options will be added to the individual items' response presentations (see above). Again, responses will be compared between participants with and without preregistration experience. Both the coding sheets and the open text inputs will be published alongside the other data once the study has been conducted. Furthermore, all analysis scripts will be included in the supporting information (see S3 Script).

**Hypotheses tests.**   In order to find out how the intention to preregister is formed, and whether research experience has an impact on attitudes and the intensity of motivations and perceived obstacles regarding preregistration, we will analyze participants' responses to the attitude, subjective norm, perceived behavioral control, intention, motivation, and obstacle scale (see S3 Script for all analyses). Only complete datasets (see section *exclusion and missing data*) will be considered for these analyses. Before the analyses, the aforementioned scales will be inspected, and the mean, standard deviation, and distribution of responses will be displayed per item in a table for easy inspection. Next, means, standard deviations, and ranges will be computed for each scale.

A significance level of $\alpha$ = .05 will be used for our hypotheses tests. Before conducting statistical analyses, assumptions will be tested for each method (see S2 Script). If assumptions are violated and tests are not robust against these violations, alternative methods will be used as described below.

For hypothesis 1, we predict that attitudes, subjective norms, and perceived behavioral control influence researchers' intention to preregister their studies in the near future. Perceived importance of the topic of preregistration will be included as moderator for attitudes, and previous preregistration experience will serve as a control variable. To test this hypothesis, a moderated multiple regression model will be computed. Assumptions for this model include linearity, uncorrelated predictors, independence and normality of residuals, and homogeneity of variance. Linearity and uncorrelated predictors are assumed on a theoretical basis. Yet, there might be a high multicollinearity for the product term (attitudes x perceived importance). This will be examined by calculating the tolerance statistics as well as the variance inflation factor. If the multicollinearity is too high for these two predictors (variance inflation factor > 10), importance will be aggregated with the attitudes scale instead of including it as a moderator, since it might be just another aspect of attitudes. In this case, the item concerning perceived importance would be included as another item in the attitude scale which would now include 25 instead of 24 items. The mean score would then be re-calculated and the new score would be included in the analyses. The independence of residuals is furthermore achieved by the survey's design. Normality of residuals and homogeneity of variance will be investigated by plotting the residuals (scatterplots, Q-Q plots and histograms). A violation of the normality of residuals is not impactful with a high *N*, as it is the case with our sample, thus a violation would not be problematic. If the assumption of homogeneity of variance is violated, weighted least squares regression will be used instead of ordinary least squares.

After testing these assumptions, the regression model will be computed: The intention (i.e., mean score of three items) will be included as dependent variable. As predictors, the attitude scale (i.e., mean score of 24 items inquiring about how positive preregistrations are perceived), perceived importance of preregistration (one item), and their product term, the subjective norm scale (i.e., mean score of eight items inquiring about how beneficial participants perceive the social norm of using preregistration), the perceived behavioral control scale (i.e., mean score of five items inquiring about perceived controllability of the behavior), and previous preregistration experience (one item: yes vs. no) will be included. We expect that higher scores on the attitude scale (i.e., more positive attitudes, see *hypothesis 1.1*), higher scores on the

subjective norm scale (i.e., higher perceived social pressure, see *hypothesis 1.4*), higher scores on the perceived behavioral control scale (i.e., higher perceived control, see *hypothesis 1.5*) are positive predictors for participants' intention scores (i.e., higher intention to preregister their studies in the near future). Additionally, we expect that perceived importance of preregistration is a positive predictor for intention (see *hypothesis 1.3*), and also significantly moderates the strength of the influence of attitudes (see *hypothesis 1.2*). Furthermore, we expect that the overall model including all predictors can significantly predict researchers' intention to preregister their studies in the future (see *hypothesis 1.6*). Because of these directional hypotheses, the regression weights will be tested in a one-tailed fashion. Standardized regression weights will be interpreted and compared. Preregistration experience will be included into the model to control for a potential sampling bias. We have no strong predictions regarding the direction of this effect, since it might be the case that a) researchers who have used preregistration before are more likely to preregister again or b) that their preregistration experiences were negative and they will less likely preregister again. Therefore, in contrast to the other predictors, this effect will be tested in a two-tailed fashion.

If all main predictors (attitudes, subjective norm, perceived behavioral control, perceived importance, and its interaction with attitudes) and the overall model are significant *(support for all hypotheses 1.1–1.6)*, we will conclude that the theory of planned behavior is indeed well suited to be used to predict researchers' intention to preregister. If the overall model, but not all predictors are significant (*support for hypothesis 1.6*, *no or partly support of hypotheses 1.1–1.5*), the theory of planned behavior will still be seen as applicable, with the significant predictors being more relevant in the context of preregistration than the non-significant ones. If the individual predictors as well as the overall model are not significant (*no support for hypotheses 1.1–1.6*), it will be concluded that the theory of planned behavior cannot be applied to the context of predicting researchers' intention to preregister, but that it might be beneficial to test it again with other open science techniques.

The second hypothesis predicts that research experience (i.e., the years someone has worked in research) influences researchers' attitudes (see *hypothesis 2.1*) as well as motivations (see *hypothesis 2.2*) and the perception of obstacles (see *hypothesis 2.3*). To investigate this, three multiple regression models will be computed: Mean scores on the attitude scale (i.e., mean score of 24 items inquiring about how positive preregistrations are perceived), motivation scale (i.e., mean score of ten items measuring how strongly participants agree with potential motivations to preregister), and obstacle scale (i.e., mean score of ten items measuring how strongly participants agree with potential obstacles to preregister) will serve as dependent variables in the respective regression models, while research experience (i.e., the years someone has worked in research) and the preregistration experience (control variable) will serve as predictors. As our hypotheses regarding the impact of research experience on attitudes, and intensity of motivations and perceived obstacles are non-directional, the regression weights will be analyzed with two-tailed tests. As three regressions will be conducted to test one set of hypotheses, a Bonferroni-Holm correction will be used to adjust the alpha level. The assumptions for these regressions will be tested in the same manner as those of the regression model for hypothesis 1 (see above).

If all three regression models show significant effects (*support for hypotheses 2.1–2.3*), we will conclude that research experience has an influence on how researchers perceive preregistration, as well as its motivations and obstacles. If only one or two of the regression models are significant (*partly support of hypotheses 2.1–2.3*), we will conclude that only for these scales, research experience has an influence, while if no significant effects can be detected (*no support for hypotheses 2.1–2.3)*, we will conclude that, against our predictions, research experience has no influence on attitudes, motivations, and perceived obstacles concerning preregistration.

## Pilot study

A pilot study was conducted to estimate the response rate and to test the recruitment method as well as the survey items. The pilot study featured mostly the same items as the main study but additional items, prompting participants for feedback about comprehensibility via open text input fields, were presented ("Did you have any problems answering the items of this page? Was anything unclear?"). Some items were adjusted based on the results. Quotas were not applied.

We invited $N = 200$ participants to partake in the pilot study (100 were invited from OSF Registries, 50 from Web of Science, and 50 from PubMed) and sent a reminder one week after the initial invitation. Participations typically occurred shortly after each email contact. The pilot study was accessible for one month overall. In this time, 29 participants (17 PhD students, three postdocs, seven professors, and two members of other academic groups which were screened out) started the survey of which 20 completed it (14 PhD students and 6 professors), yielding an overall response rate of 10%. Out of these 20, 18 (90%) participants had preregistration experience. No specific patterns in dropout behavior were found (e.g., dropping out at a certain position). No floor or ceiling effects were found. Reliability analyses for the measured scales were conducted, which showed a very high reliability for the attitude ($\alpha = 0.90$) and obstacle scale ($\alpha = 0.87$), an acceptable reliability for the motivation scale ($\alpha = 0.77$), and a moderate reliability for the subjective norm ($\alpha = 0.65$) and perceived behavioral control scale ($\alpha = 0.63$). One item of the motivation scale correlated negatively with the remaining scale items and Cronbach's alpha increased by removing it (item M2, see S1 Table). Nevertheless, the item will remain in the survey and will be checked again in the main study, since the pilot results are based on a very small sample and can only be interpreted with caution.

After conducting the pilot study, the sampling strategy was revised by extending the recruitment of participants to four instead of two general databases for psychological articles and by inviting less participants via OSF Registries (10% instead of 50%). Moreover, some comments indicated ambiguous interpretations which led us to slightly modify the wording of respective items or to change categories (i.e., academic position was replaced by academic degree in the sociodemographic part of the survey). Lastly, a few items were added, e.g., two items assessing intention following a manual about constructing questionnaires based on the theory of planned behavior [56].

## Supporting information

**S1 Text. Power analyses.** Power analyses for both hypotheses are presented here. For hypothesis 1, power analyses were performed for the entire model as well as for the individual predictors.
(DOCX)

**S2 Text. Data collection procedure.** The data collection procedure that is displayed in Fig 2 is further specified. Particularly, the specific procedures for collecting contact addresses as well as for inviting participants are described.
(DOCX)

**S1 Table. Survey items.** The items of the survey are presented. If they were derived from other studies, the original items as well as references are given.
(XLSX)

**S1 Video. Screen recording of the survey procedure (version A).** This screen recording shows the procedure of the survey for participants that responded with "yes" on the item

"Have you preregistered a study before?" (as this is a filter question for some of the following items) and who answered the knowledge check of the preregistration definition correctly.
(MP4)

**S2 Video. Screen recording of the survey procedure (version B).** This screen recording shows the procedure of the survey for participants that responded with "no" on the item "Have you preregistered a study before?" (as this is a filter question for some of the following items) and who were shown the definition of preregistration again due to errors in the knowledge check.
(MP4)

**S1 Script. Preprocessing script (preliminary).** This R script contains all steps of preprocessing. The hereby processed data will be used for the assumption tests as well as for the descriptive reports and hypotheses tests. If necessary, small changes may be made to enable correct functioning (when variables have been inserted).
(R)

**S2 Script. Assumption testing script (preliminary).** This R script will be used for testing the assumptions of the statistical models used to test hypotheses 1 and 2. If necessary, small changes may be made to enable correct functioning (when variables have been inserted).
(R)

**S3 Script. Analysis script (preliminary).** This script will be used to conduct the descriptive reports as well as the hypotheses tests of the survey. If necessary, small changes may be made to enable correct functioning (when variables have been inserted). Furthermore, analyses might be added if alternative approaches need to be used (due to assumption violation).
(R)

## Acknowledgments

We are grateful to H. Bargon and A. Kuznik for their help with testing the survey, and their future help with recruitment and coding, and also to T. Dauber and J. Pauquet for testing the survey and proofreading.

## Author Contributions

**Conceptualization:** Lisa Spitzer, Stefanie Mueller.

**Data curation:** Lisa Spitzer.

**Formal analysis:** Lisa Spitzer.

**Investigation:** Lisa Spitzer.

**Methodology:** Lisa Spitzer, Stefanie Mueller.

**Project administration:** Stefanie Mueller.

**Resources:** Stefanie Mueller.

**Software:** Lisa Spitzer.

**Supervision:** Stefanie Mueller.

**Visualization:** Lisa Spitzer.

**Writing – original draft:** Lisa Spitzer.

Writing – review & editing: Stefanie Mueller.

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
