## [Decision Letter · Decision Letter 0]

25 Sep 2020

PONE-D-20-20445

Attitudes and experiences regarding preregistration in psychological research

PLOS ONE

Dear Dr. Spitzer,

Thank you for submitting your manuscript to PLOS ONE. After careful consideration, we feel that it has merit but does not fully meet PLOS ONE’s publication criteria as it currently stands. Therefore, we invite you to submit a revised version of the manuscript that addresses the points raised during the review process.

First of all, I'd like to thank **the four reviewers that have assessed the manuscript.** The manuscript was indeed reviewed by two early career researchers, one senior researcher (from the same group of one of the ERC), one statistical reviewer and myself. I had also asked another reviewer to join us but at the end he did not answered and we are a bit late in providing feedback (I apologize for the delay in answering). Based on the comments I received, I don't think that another feedback will be useful. Reviewers made a good job, thanks. 

We all agree that it has merits and may pass the bar of in principle acceptance providing you are able to address all the major and minor issues that were raised. 

We look forward to receiving your revised manuscript.

Kind regards,

Florian Naudet, M.D., M.P.H., Ph.D.

Academic Editor

PLOS ONE

Journal Requirements:

Reviewers' comments:

Reviewer's Responses to Questions

**Comments to the Author**

1. Does the manuscript provide a valid rationale for the proposed study, with clearly identified and justified research questions?

Reviewer #1: Yes

Reviewer #2: Yes

Reviewer #3: Partly

Reviewer #4: Yes

2. Is the protocol technically sound and planned in a manner that will lead to a meaningful outcome and allow testing the stated hypotheses?

Reviewer #1: Yes

Reviewer #2: Yes

Reviewer #3: Partly

Reviewer #4: Partly

3. Is the methodology feasible and described in sufficient detail to allow the work to be replicable?

Reviewer #1: Yes

Reviewer #2: Yes

Reviewer #3: No

Reviewer #4: No

4. Have the authors described where all data underlying the findings will be made available when the study is complete?

Reviewer #1: Yes

Reviewer #2: Yes

Reviewer #3: No

Reviewer #4: No

5. Is the manuscript presented in an intelligible fashion and written in standard English?

Reviewer #1: Yes

Reviewer #2: Yes

Reviewer #3: Yes

Reviewer #4: Yes

6. Review Comments to the Author

You may also provide optional suggestions and comments to authors that they might find helpful in planning their study.

Reviewer #1: Important note: This review pertains only to ‘statistical aspects’ of the study and so ‘clinical aspects’ [like medical importance, relevance of the study, ‘clinical significance and implication(s)’ of the whole study, etc.] are to be evaluated [should be assessed] separately/independently. Further please note that any ‘statistical review’ is generally done under the assumption that (such) study specific methodological [as well as execution] issues are perfectly taken care of by the investigator(s). This review is not an exception to that and so does not cover clinical aspects {however, seldom comments are made only if those issues are intimately / scientifically related & intermingle with ‘statistical aspects’ of the study}. Agreed that ‘statistical methods’ are used as just tools here, however, they are vital part of methodology [and so should be given due importance].

COMMENTS: Your ABSTRACT is well drafted but assay type. Because your article type is ‘Registered Report Protocol’ on a very special viewpoint on preregistration of studies, I can understand that it is difficult to covert it as ‘Structured summary’. However, it will definitely be more informative if you can do that, I guess [even if your article type is ‘Registered Report Protocol’]. Please note that it is preferable to divide the ABSTRACT with small sections like ‘Objective(s)’, ‘Methods’, ‘Results’, ‘Conclusions’, etc. which is an accepted practice of most good/standard journals [including PLOS] whatever the article type may be.

Why it is not indicated in the title that this a ‘Protocol’? [in my opinion, it is better as it helps reader(s) to read the article with that perspective].

In the context of material given in lines 290-3, it may be noted that

“Whenever response options ranged from 1=strongly disagree to 4=strongly agree (or from 1=very bad to 3=neither good nor bad to 5=very good), while using a ‘Likert’ scale responses, recoding [like strongly disagree=-2, disagree=-1, neutral=0, agree=1, strongly agree=2] may yield correct and meaningful ‘arithmetic mean’ which is useful not only for comparison but has absolute meaning, in my opinion. Application of any statistical test(s) assume that meaning of entity used (mean, SD, etc) has a particular meaning. Though ‘α’ [alpha] or most other measures of reliability/correlation will remain same, however. Use of non-parametric methods should/may be preferred while dealing with data yielded by any questionnaire/score.”.

Where and how to use it will be decided by the authors. But it is given here because I thought that it may be useful [for all analyses (including ‘Confirmatory analyses’ described in lines 375-437)] proposed in this protocol. Though I am not familiar with most of the ‘software’(s) named in ‘Data analysis’ section [lines 375-429], I am confident that authors will use them appropriately.

Because the topic of this study is very interesting as well as going to be very useful, in my opinion, worth publishing (may be after minor revision).

Reviewer #2: The reviewer wants to congratulate the authors on this excellent registered report.

Not only is the study itself important and can give a clearer picture of pre-registration, the

fact that the authors are completely open with their materials is highly appreciated.

Furthermore, the authors are guiding the reader easily and clearly through their methods.

Especially the videos, guiding through the survey, show high transparency.

However, some points still need to be clarified. Thus, I give the following recommendation.

Revision with minor changes

Minor remarks:

Title: needs to be clearer. No word of survey although this represents the heart piece of the

study, registered report, quality of research

Abstract:

L 15 Immutable. Really? It is mutable and at least in OSF these dates can be tracked (you

state so yourself in l 140)

L 109: More explanation about Registered Report is missing especially as this manuscript is

one itself. The explanation seems too short, when there is a long explanation of pre-

registration. Registered Reports are a new form of pre-registration and include peer review

whereas pre-registrations don’t. Moreover, preregistration is a faster way to publish, as it

has to go through no quality check.

L 150: Wording. If you want to keep this phrase, please include “to our knowledge”

L 186 How will you cope with the different age of professors? ERC / Young professors might

be more open to the idea of Pre reg than ones who are close to retirement.

L 233: Will the search be executed by two people ?

L 233 Don’t you introduce a huge bias if you select half of the participants from OSF. I see

the discussion in L 401 but still would like to have more explanation

L 242: please put the calculation in the extra file and only put the sources, or give the min

and max values and only put the references

L 243: For the value 26.77 you put two numbers after the decimal point and this seems

strange as before you always went with one. However, probably this format has been

chosen to not change the sample size of 484.

L 268: Is excluding Master students who will not continue in Research not introducing a

bias? How do you make sure the ones who say they stay in academia the ones who will

leave do so they because of the lack of pre-registration and open science in their lab?

L 276 The survey language is English, or are there other options as well (German)?

Major remarks

Title:

Please give more information about what kind of research this is (survey + registered report)

Abstract:

I know that there is no specific guideline for registered reports. However, try to stick to one.

Like PRISMA or STROBE.

Please write more about the methods you will use. I am missing the searched databases

OSF/Web of science and that this includes an online survey.

No words about the different groups studies (PhD, Postdoc, ...) could be found.

Furthermore, the number of subjects studies is missing.

General comments:

Adding a section of Limits / Risk would be good

(e.g. is external validity given, if half of sample size already registered a protocol)

What if, if still after two rounds the quotas for professors might not be filled?

Or what if you find 175 professors who are willing to participate but not all of them will

finish the study.

What to do in case of deviations or if too many results show ceiling or floor effects.

What is the time frame of the study? Can this study be finished in one year?

I declare myself unable to assess the statistical methods used.

Reviewer #3: The authors aim to probe what students and researchers in psychology think about preregistration. They plan to do so mainly through a quantitative approach with nearly 100 clearly defined questions. Before going forward with this study as a Registered Report, I strongly recommend that the authors conduct a pilot study.

Below, I outline major comments about the design and then about the writing. I have also attached a pdf of the manuscript with minor comment boxes I made as I read the manuscript. I see much value in this study, but feel it needs to be conducted meticulously to ensure it serves the purpose it is intended for.

1. It appears that the authors want to assess attitudes and experiences from a sample that is representative of psychology researchers and students. However, because the study will recruit from the OSF and social media, I expect the sample to be biased towards those who have more experience and are more positive towards preregistration. If the authors decide to recruit from OSF and Web of Science, I recommend they identify these groups as distinct study samples and highlight that the OSF sample is unlikely to represent the average psychology researcher. Also, a 26% response rate seems unreasonably optimistic (e.g., Houtkoop et al 2018 emailed authors lists about data sharing and had a 5% response rate)

2. The specific (operationalized) variables and analyses are unclear. It’s not clear (i) what values they are going to put in their analyses, (ii) what specific (operationalized) questions the analyses are being powered for, or (iii) what conclusions will be drawn given different outcomes. I recommend itemizing each hypothesis alongside its operationalized variables and power calculations. Also, the definition of professor varies between different countries and should be clearly defined.

I feel comments 1 and 2 could be addressed through a pilot study. A pilot study would provide information on response rates, provide data that the authors could run a pilot analysis on, and potentially provide insights about sampling bias. The authors could consider adding the response option “I don’t understand this question” to the pilot. This would allow them to ensure all the questions are well understood. The authors also plan to add questions based on the responses to the first 10% of open text responses. This process could be done with the pilot data.

The following comments are on the writing.

3. The authors make several overstatements in the introduction section. I’ve identified these with comment boxes in the attached pdf.

4. Several terms need to be better defined. For example, open science and reproducibility have broad definitions, but the authors appear to be using them in a more narrow sense without explicitly stating how. Terms like ‘problematic’ and ‘insufficient’ are used without explain what the problem is or what they are insufficient for. I’ve identified these with comment boxes in the attached pdf.

5. The introduction overplays the role of psychology in open science and overlooks the influence of other fields (e.g., prospective registration in medicine, pre analysis plans in economics). I feel that the open science movement is much broader than psychology and that researcher across disciplines can benefit each other by recognizing our shared mission and best practices in each field. I don’t feel that the author’s need to justify their decision to do this study in psychology so strongly.

I always sign my reviews,

Robert Thibault (with input from my immediate working group: Jackie Thompson, Mark Gibson, and Robbie Clark).

Reviewer #4: I suggest a major revision. Please see my attached comments for more details on my suggested changes.

7. PLOS authors have the option to publish the peer review history of their article (what does this mean?). If published, this will include your full peer review and any attached files.

Reviewer #1: No

Reviewer #2: No

Reviewer #3: **Yes: **Robert Thibault

Reviewer #4: **Yes: **Katie Drax

---

## [Author Response · Author response to Decision Letter 0]

16 Feb 2021

Response to reviewers

We thank the reviewers for reading the manuscript thoroughly and for providing constructive criticism that, we believe, has resulted in a significant improvement of the manuscript. We addressed all points to our best knowledge. 

Overall, we have revised the title and short title, restructured the abstract, conducted a pilot study, shortened the introduction and removed potential overstatements, clarified the rationale of our intended sample, and reduced ambiguities in our descriptions. For this reason, we also restructured the analysis scripts slightly so that there is a preprocessing script (instead of including the preprocessing steps in the analysis script), and we describe the data analysis in more detail in the manuscript. Additionally, we described more precisely where we will share our data after conducting the study (cf. p. 18, l. 365).

Besides addressing the reviewers’ comments, we also improved wording and grammar, updated one term of the theory of planned behavior (“subjective norm” instead of “social norm”), and updated our institute’s name from “Leibniz Institute for Psychology Information” to “Leibniz Institute for Psychology” since the name was changed recently.

Please find our responses to each point below. If line numbers were given by the reviewers, we include the term, sentence, or section, to which the comment refers to, in brackets. 

Reviewer #1

Rev 1: Your ABSTRACT is well drafted but assay type. Because your article type is ‘Registered Report Protocol’ on a very special viewpoint on preregistration of studies, I can understand that it is difficult to covert it as ‘Structured summary’. However, it will definitely be more informative if you can do that, I guess [even if your article type is ‘Registered Report Protocol’]. Please note that it is preferable to divide the ABSTRACT with small sections like ‘Objective(s)’, ‘Methods’, ‘Results’, ‘Conclusions’, etc. which is an accepted practice of most good/standard journals [including PLOS] whatever the article type may be.

Thanks for the suggestion, it helped us to make our abstract much clearer. The abstract was edited to a structured abstract, using the sections “background” and “methods”. When the Stage 2 Registered Report is submitted, the sections “results” and “discussion” will be added and we may move confirmed hypotheses to the results section if the reviewers feel that this is appropriate (e.g., by changing “Based on the theory of planned behavior, we predict that positive attitudes (moderated by the perceived importance of preregistration) as well as a favorable subjective norm and higher perceived behavioral control positively influence researchers’ intention to preregister (hypothesis 1)” to “Positive attitudes (moderated by the perceived importance of preregistration) as well as a favorable subjective norm and higher perceived behavioral control positively influenced researchers’ intention to preregister” if the hypothesis is confirmed).

Rev 1: Why it is not indicated in the title that this a ‘Protocol’? [in my opinion, it is better as it helps reader(s) to read the article with that perspective].

This is our first submission of a Registered Report and we wrote the title having the final manuscript in mind. We now revised the title following the suggestion to: “Registered Report Protocol: Survey on attitudes and experiences regarding preregistration in psychological research”. Additionally, the short title was updated from “Attitudes and experiences regarding preregistration in psychology” to “Survey on preregistration in psychology”.

Rev 1: In the context of material given in lines 290-3, it may be noted that

Whenever response options ranged from 1=strongly disagree to 4=strongly agree (or from 1=very bad to 3=neither good nor bad to 5=very good), while using a ‘Likert’ scale responses, recoding [like strongly disagree=-2, disagree=-1, neutral=0, agree=1, strongly agree=2] may yield correct and meaningful ‘arithmetic mean’ which is useful not only for comparison but has absolute meaning, in my opinion. Application of any statistical test(s) assume that meaning of entity used (mean, SD, etc) has a particular meaning. Though ‘α’ [alpha] or most other measures of reliability/correlation will remain same, however. 

We changed the scales so that all scale items will be measured using a 7-point scale (not 5-point as indicated in the comment because this 1) still makes it possible to have 0 as the neutral option, 2) might differentiate better, and 3) is recommended by the author of the theory of planned behavior. Because here the middle category has absolute meaning, we have decided against centering the variables for our moderated multiple regression (which we had planned in our first draft).

Rev 1: Use of non-parametric methods should/may be preferred while dealing with data yielded by any questionnaire/score..

Although your suggestion is reasonable, we would like to stick with the metric tests as the first option to maintain the link to previous research on the theory of planned behavior. We will, however, test the assumptions of our models and resort to alternative (non-parametric) procedures in case of violations, as described in the manuscript (cf. p. 22f, l. 447 ff; p. 24 f, l. 497 ff; p. 25, l. 521 ff). For one of the analyses (planned contrasts), we furthermore edited the alternative method to a Mann–Whitney U test (cf. p. 25, l. 522 f).

Reviewer #2

Minor remarks:

Rev 2: Title: needs to be clearer. No word of survey although this represents the heart piece of the study, registered report, quality of research

[Title of first draft: “Attitudes and experiences regarding preregistration in psychological research”]

We revised the title accordingly to “Registered Report Protocol: Survey on attitudes and experiences regarding preregistration in psychological research”. Additionally, the short title was updated from “Attitudes and experiences regarding preregistration in psychology” to “Survey on preregistration in psychology”.

Abstract:

Rev 2: L 15 Immutable. Really? It is mutable and at least in OSF these dates can be tracked (you state so yourself in l 140)

[L 13 ff: “One prominent open science technique is the preregistration of studies, which comprises creating and publicly sharing a research plan that is time-stamped and immutable before the data collection of a study has begun”]

We have looked into this again: You can upload new versions of the preregistration on the OSF (this is done automatically when you add a document of the same name to the project), the old versions are then still visible in a version control. It is also possible to delete the old preregistration and upload a new version, but of course the timestamp will be a new one. Accordingly, the original version of the preregistration itself cannot be changed and is therefore “immutable”. However, there are many ways to deal with deviations by uploading new versions or adding a deviation log. Nevertheless, we have removed the description "immutable" from our manuscript to decrease ambiguity.

Rev 2: L 109: More explanation about Registered Report is missing especially as this manuscript is one itself. The explanation seems too short, when there is a long explanation of pre-registration. Registered Reports are a new form of pre-registration and include peer review whereas pre-registrations don’t. Moreover, preregistration is a faster way to publish, as it has to go through no quality check.

[L 117 ff: “Preregistration also prioritizes theorizing and methods over results as the main criterion for judging the quality of research [20] which can further decrease publication bias (this might especially be the case for reviewed preregistrations, so-called Registered Reports [28]).”]

Since the process (peer-reviewed vs. non-peer-reviewed) as well as the incentives (possibility of in-principle acceptance with Registered Reports) differ between “traditional” preregistration and Registered Reports, we believe that it is quite possible that attitudes/motivations toward Registered Reports may be different than toward preregistration. Therefore, since we want to focus on preregistration, we have decided to delete the Registered Report sections altogether so as not to confuse the readers.

Rev 2: L 150: Wording. If you want to keep this phrase, please include “to our knowledge”

[L 149 f: “... until now no systematic evaluation focusing on preregistration in psychology has been conducted.”]

We added “to our knowledge” to the sentence: “While previous surveys inquired about preregistration in the more general context of open science [22,24,35,36], to our knowledge no comprehensive study focusing on preregistration has been conducted.” (p. 6, l. 101 ff).

Rev 2: L 186 How will you cope with the different age of professors? ERC / Young professors might be more open to the idea of Pre reg than ones who are close to retirement.

[L 185 f: “Second, we want to examine how attitudes, motivations and perceived obstacles differ between master students, PhD students, post-docs, and professors.”]

We now ask participants to indicate their age, and we will analyze exploratory if the age has any influence. However, since we do not describe any exploratory analyses in the Registered Report Protocol, we do not describe this option in the manuscript. Furthermore, we also added an open text input for "professor types" so that participants have more flexibility in indicating their position (e.g., junior professor).

Rev 2: L 233: Will the search be executed by two people ?

[L 233 ff: “To recruit this sample, a method similar to the one by Field et al. [62] will be used. The term “psychology” will be searched for on OSF Registries (https://osf.io/registries) and Web of Science (https://apps.webofknowledge.com/) and resulting documents will be sorted from newest to oldest. These documents will be scanned for authors whose e-mail addresses will then be identified, not only including the first or corresponding author but all authors for whom contact information can be found via personal websites linked to OSF or Web of Science, or by searching for the author via Google, Google Scholar, Twitter, and ResearchGate. Detailed information about this procedure is given in the supplementary material (see S2 Text).”]

Three persons will conduct this search in a joint effort (a fourth might be included if the workload is too high).

Rev 2: L 233 Don’t you introduce a huge bias if you select half of the participants from OSF. I see the discussion in L 401 but still would like to have more explanation

[L 233 ff: “To recruit this sample, a method similar to the one by Field et al. [62] will be used. The term “psychology” will be searched for on OSF Registries (https://osf.io/registries) and Web of Science (https://apps.webofknowledge.com/) ...”]

We added the following paragraph to explain our rationale for this decision:

“This recruitment procedure might result in a sample biased toward researchers who have already preregistered (see section pilot study). However, this is not an issue for our survey because the main focus does not lie on gaining a representative sample of all psychological researchers, but on identifying problems in the current implementation of preregistration and uncovering reasons for psychological researchers’ potential ambivalence toward it.” (p. 13, l. 244 ff).

Additionally, we added PubMed as another source (1/2 of participants are recruited from OSF, 1/4 from WoS, and 1/4 from PubMed, cf. p. 9, l. 171 ff) to get a more diverse sample (however, this will most likely not result in a sample without this bias).

Rev 2: L 242: please put the calculation in the extra file and only put the sources, or give the min and max values and only put the references

[L 241 f: “Response rates were rather low in many studies that focused on researchers as sample (e.g., 36.1% [5]; 36.1% [34]; 8.2% [45]; 45.9% [46]; 6% [62]; 17.6% [63]; 37.5% [64]).”]

As suggested by reviewers #3 and #4, we conducted a pilot study and now base our estimate of the response rate on the results of the pilot (cf. p. 13, l. 251 f). Thus, we deleted the calculations.

Rev 2: L 243: For the value 26.77 you put two numbers after the decimal point and this seems strange as before you always went with one. However, probably this format has been chosen to not change the sample size of 484.

[L 242 f: “Based on these studies, the response rate of our survey is estimated to be around 26.77% (mean of the above).”]

This is no longer applicable because we revised the expected response rate (cf. p. 13, l. 251 f).

Rev 2: L 268: Is excluding Master students who will not continue in Research not introducing a bias? How do you make sure the ones who say they stay in academia the ones who will leave do so they because of the lack of pre-registration and open science in their lab?

[L 264 ff: “All participants that indicate that their research does not fall within the scope of psychology (or for master students, whose major is not psychology), or indicate that they are neither master student, PhD student, post-doc nor professor will be screened out at the beginning of the survey, i.e., they will be directed to an exit page rather than to the main body of the survey. The same will apply for master students that indicate that they do not intend pursuing an academic career, as the survey targets a research oriented population.”]

Examining the reasons why master students decide to leave academia exceeds the scope of our survey as there could be many (e.g., in Germany, most psychology students continue to become therapists rather than pursuing a career in research). We decided to restrict our sample to researchers for whose work preregistration is directly relevant. That excludes anyone who leaves academia, irrespective of their reasons.

Rev 2: L 276 The survey language is English, or are there other options as well (German)?

[L 276 f: “The online survey was created with the software SoSci Survey (version 3.2.06) [65] and will be supplied via www.soscisurvey.de.”]

Yes, the survey will only be presented in English and we added this information to the manuscript (cf. p. 15, l. 295).

Major remarks:

Title: 

Rev 2: Please give more information about what kind of research this is (survey + registered report)

We revised the title accordingly to “Registered Report Protocol: Survey on attitudes and experiences regarding preregistration in psychological research”. Additionally, the short title was updated from “Attitudes and experiences regarding preregistration in psychology” to “Survey on preregistration in psychology”.

Abstract:

Rev 2: I know that there is no specific guideline for registered reports. However, try to stick to one. Like PRISMA or STROBE.

We have used the PRISMA guidelines as orientation and used the sections “background” and “methods”. The sections “results” and “discussion” will then be added in the Stage 2 Registered Report.

Rev 2: Please write more about the methods you will use. I am missing the searched databases OSF/Web of science and that this includes an online survey. 

No words about the different groups studies (PhD, Postdoc, ...) could be found.

Furthermore, the number of subjects studies is missing.

We revised the abstract by adding the requested information. We now write that we conduct an online survey, use a mixed-methods approach, how participants will be recruited, what groups will be investigated and what the targeted N is.

General comments:

Rev 2: Adding a section of Limits / Risk would be good

(e.g. is external validity given, if half of sample size already registered a protocol)

We now mention potential limitations (i.e., potential bias due to the survey method, possible violation of the assumptions of our statistical models) in the manuscript (cf. p. 13, l. 244 ff; p. 22, l. 440 ff). We will further discuss limitations that may arise during the data collection or data analyses in the discussion section if we submit the study as a Registered Report Research Article.

Rev 2: What if, if still after two rounds the quotas for professors might not be filled?

Or what if you find 175 professors who are willing to participate but not all of them will finish the study.

It would not be a problem for the quota counting if members of one academic group are willing to participate but drop out, since the counting is performed on the last page of the survey, not when entering the survey (so drop-outs will not influence the quotas). Results of the pilot study suggest that professors are willing to participate (of overall N = 29 who started the pilot study, seven were professors, and six completed it) which leaves us hopeful. But even if the quota cannot be filled, we defined all quotas for a slightly bigger sample than the necessary N so that it is possible that e.g., more PhD students and postdocs can participate to compensate for this (this would then result in a slightly higher percentage of these groups, but it will be ensured that not all participants belong to the same group, cf. p. 12, l. 233 ff).

Rev 2: What to do in case of deviations or if too many results show ceiling or floor effects.

We will check for floor/ceiling effects of individual survey items and will exclude items for which ≥90% of participants indicated the lowest/highest category. So far, we found no floor and ceiling effects in the pilot study. Furthermore, we will report any deviation from the preregistered procedure.

Rev 2: What is the time frame of the study? Can this study be finished in one year?

We started to collect email addresses already in order to be able to finish the study within one year. We also added this information to the manuscript: "The collection of email addresses has started on January 4th, 2021. Emails will be sent once our submission receives an in-principle-acceptance and 4840 email addresses have been collected using the procedure described above." (p. 14, l. 268 ff).

 

Reviewer #3

Rev 3: Before going forward with this study as a Registered Report, I strongly recommend that the authors conduct a pilot study.

We are very grateful for the reviewer’s suggestion to conduct a pilot study, which we have done prior to re-submitting the manuscript. The insights gained from this pilot study have been very beneficial (see new manuscript section pilot study, p. 26 f).

Since participants in the pilot took a little longer than we had expected to complete the survey, we deleted one ranking item (“What do you think is the most important information to preregister?”) to make the survey a bit shorter. Additionally, we now estimate that the survey takes 20-25 minutes instead of 20 minutes. 

Furthermore, we updated the timeframe of our survey’s data collection from approximately two months to one month, since it became clear in the pilot that participants only participated shortly after being invited. We have described so in the section data collection and stopping rule (p. 12 f) and in S2 Text.

Rev 3: It appears that the authors want to assess attitudes and experiences from a sample that is representative of psychology researchers and students. However, because the study will recruit from the OSF and social media, I expect the sample to be biased towards those who have more experience and are more positive towards preregistration. If the authors decide to recruit from OSF and Web of Science, I recommend they identify these groups as distinct study samples and highlight that the OSF sample is unlikely to represent the average psychology researcher. Also, a 26% response rate seems unreasonably optimistic (e.g., Houtkoop et al 2018 emailed authors lists about data sharing and had a 5% response rate)

We agree that recruiting from the OSF will most likely not result in a representative sample. We thoroughly considered your point and revised the manuscript to communicate this limit clearly when describing our research goal (cf. p. 13, l. 244 ff). As suggested, we will track the source from which each participant was invited (via distributing different survey links) and will analyze differences between the groups exploratorily. Additionally, we now include PubMed as a further resource for inviting participants. Regarding the response rate, we corrected the estimated response rate based on the pilot study, and now expect a response rate of 10% (cf. p. 13, l. 251 f).

Rev 3: The specific (operationalized) variables and analyses are unclear. It’s not clear (i) what values they are going to put in their analyses, (ii) what specific (operationalized) questions the analyses are being powered for, or (iii) what conclusions will be drawn given different outcomes. I recommend itemizing each hypothesis alongside its operationalized variables and power calculations.

We itemized the research questions (cf. p. 6, l. 115; p. 8, l. 151 f) and hypotheses (cf. p. 7 f, l. 137 ff; p. 9, l. 158 ff) and referred to them in the power analysis and when describing the hypotheses tests. Furthermore, we added paragraphs about conclusions that will be drawn based on different patterns of outcomes (cf. p. 23 f, l. 481 ff; p. 25 f, l. 524 ff).

Rev 3: Also, the definition of professor varies between different countries and should be clearly defined.

We included an open text input in the professor item so that participants can give further comments on which type of professor they are. As also suggested by reviewer #2, we will keep this information to operationalize differences between „young“ (mainly with respect to their professional age) and „old“ professors in an exploratory way.

Rev 3: I feel comments 1 and 2 could be addressed through a pilot study (see 3 comments above)

A pilot study would provide information on response rates, provide data that the authors could run a pilot analysis on, and potentially provide insights about sampling bias. The authors could consider adding the response option “I don’t understand this question” to the pilot. This would allow them to ensure all the questions are well understood. The authors also plan to add questions based on the responses to the first 10% of open text responses. This process could be done with the pilot data.

We conducted a pilot study as suggested and describe the results in the new section pilot study (p. 26 f) in the manuscript. The item "Did you have any problems answering the items of this page? Was anything unclear?" was included in the pilot and some items were revised based on participants' comments. Furthermore, we analyzed the open text inputs and extracted coding categories for the main study based on mentioned themes (yet since there were only few responses, we will additionally inspect the first 10% of responses from the main study to generate coding categories). As suggested by Albers and Lakens (2018), we would like to implement this pilot study as an internal pilot, i.e., we would like to include the pilot data in the main survey’s analyses (cf. p. 26, l. 548 ff).

Rev 3: The authors make several overstatements in the introduction section. I’ve identified these with comment boxes in the attached pdf.

We have revised the introduction and removed the potential overstatements.

Rev 3: Several terms need to be better defined. For example, open science and reproducibility have broad definitions, but the authors appear to be using them in a more narrow sense without explicitly stating how. Terms like ‘problematic’ and ‘insufficient’ are used without explain what the problem is or what they are insufficient for. I’ve identified these with comment boxes in the attached pdf.

We included a definition for replicability (p. 3, l. 35 f), and singled out preregistration as a specific open science practice rather than referring to open science when we mean preregistration. To reduce any left ambivalence concerning the term “open science”, we included a reference to an overview of open science techniques (“A manifesto for reproducible science”, Munafò et al., 2017, doi:10.1038/s41562-016-0021, cf. p. 4, l. 54). Overall, we re-read and revised or removed all instances that reviewer #3 pointed out in the manuscript and think that it now reads much clearer. 

Rev 3: The introduction overplays the role of psychology in open science and overlooks the influence of other fields (e.g., prospective registration in medicine, pre analysis plans in economics). I feel that the open science movement is much broader than psychology and that researcher across disciplines can benefit each other by recognizing our shared mission and best practices in each field. I don’t feel that the author’s need to justify their decision to do this study in psychology so strongly.

It was not our intention to overlook the contributions of other fields to open science. Rather, as psychologists, we experienced the impact that the replication crisis had/has on our field and wanted to acknowledge the movement that emerged as a response, that is, the positive developments driven by psychologists that occurred as a consequence. We revised the introduction so that the role of psychology as a driving force of open science is less emphasized as before but instead, more weight is placed on the impact that the replication crisis had on that field and how the discipline reacted, in particular with respect to the preregistration of psychological studies. Specifically, we aim to identify reasons for the low rate of preregistration in psychology (as compared to other fields such as medicine).

Rev 3: L 17 prospective registration has been around in medicine for decades and is mandated by journals and regulatory agencies (eg, US FDA) for over a decade. I recommend you don't overstate psychology's role.

[L 17 f: “In particular, the focus will be on psychological science, as it has been a trailblazer to the open science development.”]

See the comment above.

Rev 3: L 18 Did you do a search to confirm this is true? Are there semi-systematic evaluations? Or is this 'to the best of your knowledge'?

[L 18 ff: “Until now, no systematic evaluation of current attitudes and experiences regarding preregistration in psychology, as well as motivations and perceived obstacles has been conducted.”]

We already did a thorough literature research before but checked the literature again and added some references (e.g., p. 5 f, l. 73 f).

Furthermore, the sentence this comment was referencing was deleted from the abstract, and later in the manuscript when mentioning that “no comprehensive study focusing on preregistration has been conducted”, we added “to our knowledge”: “While previous surveys inquired about preregistration in the more general context of open science [22,24,35,36], to our knowledge no comprehensive study focusing on preregistration has been conducted.” (p. 6, l. 101 ff).

Rev 3: L 23 Is this a hypothesis? If so, state so clearly.

[L 22 ff: “Based on the theory of planned behavior, we reason that attitudes (moderated by the perceived importance of the topic of preregistration) as well as social norms and perceived behavioral control influence researchers’ intention to preregister their studies.”]

The two hypotheses are now clearly enumerated and described (cf. p. 7 f, l. 137 ff; p. 9, l. 158 ff).

Rev 3: L 28 In what direction and magnitude?

[L 26 ff: “As early career researchers are oftentimes seen as the driving force of the open science movement, while senior researchers are reported to be more reluctant, we expect differences in attitudes and experiences between these groups.”]

For the first hypothesis, we wrote the expected direction of the regression weights, and for the second hypothesis we wrote that the hypothesis is non-directional, but did not explicitly say anything about the magnitude as we have no particular expectation regarding the magnitude. We will, however, report obtained effect sizes in the final manuscript.

Rev 3: L 30 The last few sentences of the abstract are vague. Are these confirmatory tests you will run or explorations. Do you have any hypotheses? If yes, state them. If no, state that you don't have any.

[L 22 ff: “Based on the theory of planned behavior, we reason that attitudes (moderated by the perceived importance of the topic of preregistration) as well as social norms and perceived behavioral control influence researchers’ intention to preregister their studies. Furthermore, we plan to investigate differences between academic groups: As early career researchers are oftentimes seen as the driving force of the open science movement, while senior researchers are reported to be more reluctant, we expect differences in attitudes and experiences between these groups. By conducting this survey, we want to examine current attitudes and experiences of psychological researchers, in order to apply this knowledge to foster the practice of preregistration.”]

We now describe the methods and planned hypotheses tests in more detail, also in the abstract.

Rev 3: L 34 potential overstatement.

[L 33 f: “... the reliability of published research findings has been a subject of discussion across all scientific disciplines.”]

We used “many” instead of “all”: “... the reliability of published research findings has been a subject of discussion across many scientific disciplines.” (p.3, l. 34 f).

Rev 3: L 35 “interdisciplinary”

[L 35: “Summarizing these concerns, the term reproducibility crisis became known interdisciplinary [3].”]

We used the term “arose” instead of “became known interdisciplinary” in the sentence “Summarizing these concerns, the term replication crisis [3] arose ...” (p. 3, l. 35).

Rev 3: L40 It appears that you just jumped from low reproducibility to false positives. They are not synonymous.

[L 40 f: “Various reasons have been discussed for the high rate of false positive research results, and are oftentimes referred to as “questionable research practices” (e.g., see [4,5]).”]

We differentiated the two terms by writing “It has been reasoned that false positives, i.e., effects that are significant in studies but do not exist in reality, contribute to low replicability [7].” (p. 3, l. 44 f).

Rev 3: L 46 problematic how?

[L 45 f: “Low statistical power (e.g., due to small sample sizes) or poor quality control (e.g., no manipulation checks) are also problematic [6].”]

This section was deleted to shorten the introduction (based on reviewer #4’s suggestion), so the comment is no longer applicable.

Rev 3: L 50 These what?

[L 50 f: “These are further associated with an insufficient distinction between prediction (confirmatory analyses) and post-diction (exploratory analyses) [9,10].”]

This section was deleted to shorten the introduction (based on reviewer #4’s suggestion), so the comment is no longer applicable.

Rev 3: L 50 insufficient for what?

[L 50 f: “These are further associated with an insufficient distinction between prediction (confirmatory analyses) and post-diction (exploratory analyses) [9,10].”]

This section was deleted to shorten the introduction (based on reviewer #4’s suggestion), so the comment is no longer applicable.

Rev 3: L 55 overstatement. I suggest rewording.

[L 55 f: “In academia, researchers are presented with the alternatives to either “publish or perish” (e.g., see [12]).”]

This section was deleted to shorten the introduction (based on reviewer #4’s suggestion), so the comment is no longer applicable.

Rev 3: L 58 overstatement

[L 56 ff: “As the likelihood of a study being accepted by a journal is higher if the obtained results are novel and confirm the hypotheses – the so-called publication bias [6] – the incentive is high to (maybe unconsciously) alter the study so that it fits the required scheme.”]

This section was deleted to shorten the introduction (based on reviewer #4’s suggestion), so the comment is no longer applicable.

Rev 3: L 64 can you define how you are using this term?

[L 64: “reproducibility”]

Since “replication/replicability” is less ambiguous than “reproducibility” and fits our situation better, we changed the wording and clearly defined replicability at the beginning of the introduction, so that the construct is used consistently and unambiguously (cf. p. 3, l. 35 f).

Rev 3: L 71 Preprints were the reason the Web was invented by physicists 30 years ago. Economists promote push-button replication and pre-analysis plans. Geneticist have large-scale collaborations with shared datasets. I recommend defining what you mean by 'open science' (e.g., does your definition include participation of minorities in the production and participation of research). And that you don't over-hype psychology when other fields have important contributions too.

[L 70 ff: “On the other hand, due to this fact, psychological science has taken on a pioneering role in developing and implementing open science techniques [6,13,15].”]

Since we focus on preregistration as one specific technique, we have shortened the overall description of open science. Instead, as described above, we included a reference to an overview of open science techniques (“A manifesto for reproducible science”, Munafò et al., 2017, doi:10.1038/s41562-016-0021) to illustrate how we use the term (cf. p. 4, l. 54). Furthermore, we emphasized the role of psychology for open science less.

Rev 3: L 75 I don't think you need to emphasize this reasoning so strongly.

[L 75 f: “Because of psychology’s impact on open science, the present survey focuses on psychologists’ views on preregistration as one prominent open science technique.”]

We edited the introduction accordingly.

Rev 3: L 79 not true for all work (eg, epidemiological datasets). 

[L 79 f: “A preregistration is a research plan that is time-stamped, immutable, created before the data collection of a study has begun ...”]

We edited the sentence to “A preregistration is a research plan that is time-stamped, created before the data has been collected or examined ...” (p. 4, l. 54 ff).

Rev 3: L 81 but not always immediately, some preregistrations are embargoed.

[L 80 f: “... most often submitted to a public registry, thus making planned study details available to others [11,19,20].”]

We added “(possibly after an embargo period)” to the sentence: “A preregistration is a research plan that is time-stamped, created before the data has been collected or examined, and most often submitted to a public registry, thus making planned study details available to others (possibly after an embargo period) [14,15].” (p. 4, l. 54 ff).

Rev 3: L 89 The idea of a file drawer has been questioned. It's unclear if the file drawer exists, or if people just hack their results into something publishable.

[L 89 ff: “Publication bias leads to the impression that there is stronger evidence for a phenomenon than actually exists since positive findings are represented disproportionately, whereas negative findings end up in a “file drawer” [6].”]

This section was deleted to shorten the introduction (based on reviewer #4’s suggestion), so the comment is no longer applicable (however, we think that this perspective is quite interesting).

Rev 3: L 93 There are also many routes one might take regardless of whether they get statistically significant results.

[L 93 f: “Instead, there are many possible routes one might take to get a significant result (referred to as “garden of forking paths” [25]).”]

This section was deleted to shorten the introduction (based on reviewer #4’s suggestion), so the comment is no longer applicable.

Rev 3: L 96 I don't think this is what NHST does. I believe it only speaks to the null hypothesis.

[L 94 ff: “When using null hypothesis significance testing (which is very prominent in various disciplines [11]), most often α = .05 is used as significance threshold to indicate that the given data is rather unlikely 96 under the null hypothesis, and therefore there is sufficient evidence for the alternative hypothesis and thus for one’s predictions.”]

This section was deleted to shorten the introduction (based on reviewer #4’s suggestion), so the comment is no longer applicable.

Rev 3: L 99 how?

[L 99: “It is problematic if the data influences statistical decisions ...”]

This section was deleted to shorten the introduction (based on reviewer #4’s suggestion), so the comment is no longer applicable.

Rev 3: L 106 which would make it easier to detect pub bias. But I don't see how this would in itself reduce pub bias.

[L 105 f: “Publication bias might be decreased by preregistrations because they are publicly available and thus discoverable (see [20]).”]

This section was deleted to shorten the introduction (based on reviewer #4’s suggestion), so the comment is no longer applicable.

Rev 3: L 109 that's what they are called. 

[L 109: “... so-called Registered Reports [28]).”]

This section was deleted to shorten the introduction (based on reviewer #4’s suggestion), so the comment is no longer applicable.

Rev 3: L 109 ...for confirmatory tests. - there is still infinite analytic flexibility for exploratory tests.

[L 109 ff: “Analytical flexibility is decreased by defining and documenting planned analyses beforehand because this way one can clearly distinguish confirmatory from exploratory analyses [9,11,20,29].”]

This section was deleted to shorten the introduction (based on reviewer #4’s suggestion), so the comment is no longer applicable.

Rev 3: L 122 It's still no where near standard.

[L 122: “Yet, it [preregistration] is still not completely common. ”]

We changed the descriptions of this situation to “is still not widely practiced in psychology” (p. 4, l. 66) and “Yet, looking at the fraction of published studies that were actually preregistered paints a different picture.” (p. 5, l. 75 f). The latter references the paragraph before which describes that a lot of psychological researchers indicated to have tried preregistration.

Rev 3: L 126 This sentence is confusing.

[L 125 f: “... they deemed many open science practices necessary, yet towards preregistration compliance decreased ...”]

We changed the sentence to “yet toward preregistration they expressed more reluctance” (p. 5, l. 80).

Rev 3: L 141 Also Ofosu and Posner 2019 showed this in economics. And Li 2018 did a meta-analysis of this issue in medicine.

[L 139 ff: “Moreover, some studies found problems with the current implementation of preregistration such as poor disclosure of deviations from preregistered plans in finished manuscripts [42].”]

Thank you for pointing this out. We included the two references in the manuscript (cf. p. 5, l. 93 ff).

Rev 3: L 152 It's not yet clear if this study is going to be quantitative or qualitative or mixed-methods. Can you state this in the abstract?

[L 152 f: “We want to examine the following questions: How do psychological researchers think of preregistration?”]

We included a description that the survey will be “mixed-methods” in the abstract and also when describing the aim of the study (cf. p. 6, l. 112 f).

Rev 3: L 164 has this been tested in terms of preregistration or other open science practices?

[L 164 f: “In our specific context, this implies that the intention to preregister predicts whether researchers will actually preregister their studies or not.”]

We added “To our knowledge, this has not yet been studied in the context of preregistration or open science.” (p. 7, l. 118 f).

Rev 3: L 182 Is this the main hypothesize? Can you operationalize each aspect of it?

[L 182 ff: “Overall, we hypothesize that attitudes (moderated by the perceived importance of preregistration), social norms and perceived behavioral control can significantly predict researchers’ intention to preregister their studies in the near future.”]

There are two main hypotheses. We itemized them to make them clearer (cf. p. 7 f, l. 137 ff; p. 9, l. 158 ff), and also referenced them in the following parts of the manuscript to increase the clarity.

Rev 3: L 184 This paragraph gets quite confusing for me. Can you itemize your hypotheses and questions?

[L 161 ff: “Besides, we want to investigate two specific research questions: First, we want to examine which factors promote or prevent preregistration. According to the theory of planned behavior [48,49], the intention to perform a behavior can be seen as a direct antecedent of the actual behavior. In our specific context, this implies that the intention to preregister predicts whether researchers will actually preregister their studies or not. Because of this relevance, we want to investigate how such intentions are formed. As described by Ajzen and colleagues [48,49], three aspects influence intentions: 1) attitudes towards the behavior (which result from the ratio of perceived advantages to disadvantages of performing the behavior), 2) the subjective norm (which represents the perceived social pressure to perform or not perform the behavior), and 3) the perceived behavioral control (which focuses on the question if the subject has the resources and skills to perform the behavior or not; also see [50–55] for meta-analytical support of this model, and [56] for an overview). We will measure attitudes towards preregistration as well as subjective norms and perceived behavioral control, and investigate how they influence researchers’ intention to preregister their studies in the future. Based on the model’s postulations, we expect that more favorable attitudes and subjective norms as well as higher perceived behavioral control positively influence the intention to use preregistration. As the relative importance of attitudes, social norms and perceived behavioral control differs in dependence of considered behaviors, situations and populations [48,49], we want to test which of these is the strongest predictor for the intention to use preregistration. Additionally, we will also include the perceived importance of preregistration as moderator for the strength of the influence of attitudes on intention. Such an extension of the model will compensate for potential non-attitudes (e.g., see [57]) and is explicitly allowed by the theory of planned behavior [48,56]. Overall, we hypothesize that attitudes (moderated by the perceived importance of preregistration), social norms and perceived behavioral control can significantly predict researchers’ intention to preregister their studies in the near future.”]

We itemized the hypotheses (cf. p. 7 f, l. 137 ff; p. 9, l. 158 ff) and furthermore added “research question 1/2” to the respective questions (cf. p. 6, l. 115; p. 8, l. 151 f).

Rev 3: L 187 qualitatively, quantitatively? Using descriptive or inferential statistics?

[L 186 f: “We predict an overall difference between these academic groups.”]

We made it clearer that we will compare how strongly motivations and obstacles are perceived and also explicitly said that we expect an “overall quantitative difference” (p. 9, l. 162). Additionally, we described which inferential models we will use when presenting the hypotheses (cf. p. 8, l. 138 ff; p. 9, l. 159 ff).

Rev 3: L 187 how do you define/operationalize these?

[L 187 f: “Moreover, we want to further inspect differences between early career researchers and more senior researchers.”]

We edited this sentence to “Moreover, we want to further inspect differences between early career researchers who we define as PhD students, and more senior researchers who we operationalize as postdocs and professors.” (p. 8, l. 152 ff).

Rev 3: L 197 I don't know what this means.

[L 196 f: “As the focus will be on preregistration in psychology, data from psychological master students, PhD students, post-docs and professors will be collected (differentiation based on [45]).”]

We deleted the reference (new sentence: “... data from psychological master students, PhD students, postdocs, and professors will be collected.”, p. 10, l. 179 f) because the description is clear without it (and the differentiation differs slightly from this source anyway).

Rev 3: L 202 What test are you powering for? This information has little meaning unless it is paired with the test you are doing.

[L 198 ff: “The sample size of N = 484 has been determined by using G*Power [58,59] in combination with a thorough review of the existing literature as described in the following paragraphs. The power analysis was specified to achieve a statistical power of 95% at a given significance threshold of 5% and a type II error probability of also 5% (α = β = .05). The here reported power analyses are available as supplementary material (see S1 Text).”]

We included a new figure that shows the specific tests and data that are used as the basis for the power analyses (see Fig 1).

Rev 3: L 217 This paragraph has little meaning to me because inferential tests are being powered for, but I don't yet know what data is being collected, or what tests are going to be done.

[L 203 ff: “To test which factors influence the intention to preregister (hypothesis 1), a multiple moderated regression model will be computed, based on the rationale of the theory of planned behavior [48,49,56]. Various meta-analyses have been conducted regarding this model, focusing on different topics (e.g., health behavior). In these, the percentage of variance of intention explained by attitudes, social norm and perceived behavioral control combined ranged between 30.4% < R² < 44.3% [50,52–55], from which the lowest reported effect size (R² = 30.4%) was chosen as minimal effect size of interest. The power analysis for the overall regression model (including all five predictors: attitudes, perceived importance of preregistration, attitudes x importance, social norm, perceived behavioral control) yielded an optimal sample size of N = 52 to achieve α = β = .05. Additional power analyses were conducted to compute the optimal sample size to test the individual predictors. As comparable effect sizes, R² based on the averaged correlations of individual variables were searched for in the afore-mentioned meta-analyses, and the smallest ones were chosen for each power analysis. This resulted in an optimal sample size of N = 42 for testing attitudes, N = 85 for testing social norms, and N = 236 for testing perceived behavioral control as predictors for intention (with α = β = .05).”]

A figure was added to display the according details (see Fig 1).

Rev 3: L 223 It appears you are powering to detect an overall difference between multiple groups, rather than to identify specific difference. Generally, I believe it makes more sense to power a study for the final post-hoc tests you are planning.

[L 218 ff: “Differences between academic groups (hypothesis 2) will be tested in a MANOVA. For this power analysis, the smallest reported effect size for group differences by Abele-Brehm et al. (η² = .05, [45] p. 4) was implemented as comparable effect size, and the lower bound of the 80% confidence interval was included as minimal effect size of interest (as recommended by [60,61]). This resulted in an optimal sample size of N = 484 to obtain α = β = .05. As this is the highest computed necessary N, it will be used as overall optimal sample size.”]

We conducted an additional power analysis for the planned contrasts (cf. p. 11 f, l. 217 ff), but as the optimal sample size for the overall MANOVA is higher, this will be used for the overall optimal sample size which will therefore remain N = 484.

Rev 3: L 228 How are you going to deal with sampling bias (e.g., those who are more keen to preregister participate)?

[L 225 ff: “To ensure that all four groups (master students, PhD students, post-docs, and professors) are nearly equally represented in our final sample, a quota sampling will be used with a 25% quota for each subgroup. Yet, as it might be the case that some quotas might not be filled (e.g., that only few professors participate), quotas are defined for a slightly bigger sample than the necessary N = 484 to compensate for this eventuality.”]

We added a paragraph to explain why a biased sample would not be an issue:

“This recruitment procedure might result in a sample biased toward researchers who have already preregistered (see section pilot study). However, this is not an issue for our survey because the main focus does not lie on gaining a representative sample of all psychological researchers, but on identifying problems in the current implementation of preregistration and uncovering reasons for psychological researchers’ potential ambivalence toward it.” (p. 13, l. 244 ff).

Rev 3: L 232 shouldn't it be 50% students (25 master, 25 phd)?

[L 230 ff: “This enables us to reach the a priori computed power even if not all quotas can be filled, while still ensuring that no group is overrepresented (e.g., it won’t be the case that 50% of the sample are students).”]

We deleted the part of the sentence “(e.g., it won’t be the case that 50% of the sample are students)” because it is not necessary (the description is clear without it) and it is misleading (we were referring to master students, but did not consider PhD students in this example).

Rev 3: L 234 Searching the osf for people to contact will lead to sampling bias. If this is the sample you won't, no prob. But if you're looking for a representative sample from psychology researchers, I wouldn't use the osf.

[L 233 ff: “To recruit this sample, a method similar to the one by Field et al. [62] will be used. The term “psychology” will be searched for on OSF Registries (https://osf.io/registries) and Web of Science (https://apps.webofknowledge.com/) ...”]

We added a paragraph to explain why a biased sample would not be an issue (see above, cf. p. 13, l. 244 ff).

Rev 3: L 239 these will all lead to sampling bias towards 'open' researchers.

[L 236 ff: “These documents will be scanned for authors whose e-mail addresses will then be identified, not only including the first or corresponding author but all authors for whom contact information can be found via personal websites linked to OSF or Web of Science, or by searching for the author via Google, Google Scholar, Twitter, and ResearchGate.”]

We added a paragraph to explain why a biased sample would not be an issue (see above, cf. p. 13, l. 244 ff).

Rev 3: L 254 sampling bias again.

[L 253 ff: “Additionally, the survey will be advertised on social media (Facebook, Twitter) and via student and researcher specific mailing lists, which will be seen as additional sources of participants.”]

We added a paragraph to explain why a biased sample would not be an issue (see above, cf. p. 13, l. 244 ff).

Rev 3: L 273 Will you analyze data from participants who only completed part of the survey?

[L 270 ff: “At the end of the survey, participants will be asked whether they responded faithfully and participants who did not will also be excluded from the analyses. All remaining participants that completed all pages of the survey will be included in the final sample. Only these participants will be counted for fulfillment of quotas and planned sample size.”]

We will only consider complete datasets for the hypotheses tests, but will also include incomplete datasets for our descriptive reports. This is explained in detail in the manuscript in the section exclusion and missing data (p. 14 f).

Rev 3: L 357 how?

[L 356 f: “First, words that were used very often will be analyzed quantitatively.”]

This analysis was removed, instead we will focus on the qualitative approach which we described in more detail (cf. 20 f, l. 413 ff).

Rev 3: L 358 themes?

[L 357 f: “Then in a more qualitative approach, coders will read the first 10% of responses and identify categories mentioned by the participants.”]

We revised this term to “categories of themes” (new sentence: “Specifically, open text inputs of each item will be shuffled and the first 10% of these shuffled responses will be used to establish initial categories of themes.”, p. 21, l. 418 ff).

Rev 3: L 364 what types of plots and tables?

[L 364 f: “Results will be displayed as plots and tables to enable easy visual inspection of the frequencies for our report’s readers.”]

We decided to use frequency tables as a consistent reporting method, whenever possible, and described so in the manuscript.

Rev 3: L 379 I see this as 3 seperate hypotheses.

[L 379 f: “For hypothesis 1, we predict that attitudes, social norms and perceived behavioral control influence researchers’ intention to preregister their studies in the near future.”]

We agree and have itemized the hypotheses and created several sub-hypotheses to reflect this (cf. p. 7 f, l. 137 ff; p. 9, l. 158 ff).

Rev 3: L 405 I believe it's best to do a two-tailed test unless the other tail is an impossibility.

[L 404 f: “Because of these directional hypotheses, the regression weights will be tested in a one-tailed fashion.”]

Based on the theory of planned behavior, we have very specific predictions regarding the direction of the effects of the first hypothesis. We predict that favorable attitudes (= higher score on attitude scale), favorable subjective norm (= higher score on SN scale), higher perceived behavioral control (= higher score on PBC scale), perceived importance (higher score on the importance item) as well as the interaction attitudes x importance are positive predictors for the intention to preregister. Thus, we would like to keep the tests one-sided (also see Greenland et al. 2016, p. 342: "One should always use two-sided P values. No! Two-sided P values are designed to test hypotheses that the targeted effect measure equals a specific value (e.g., zero), and is neither above nor below this value. When, however, the test hypothesis of scientific or practical interest is a one-sided (dividing) hypothesis, a one-sided P value is appropriate. For example, consider the practical question of whether a new drug is at least as good as the standard drug for increasing survival time. This question is one-sided, so testing this hypothesis calls for a one-sided P value. Nonetheless, because two-sided P values are the usual default, it will be important to note when and why a one-sided P value is being used instead."). However, when comparing the different academic groups (hypothesis 2), two-tailed tests will be used because in this case, deviations in both directions are not unlikely and would be relevant. 

Rev 3: Will you be getting a summary statistic for each group of variables? By summing them? how?

When describing the material we added a description of how the scales will be included in the analyses (mean scores): “For the statistical analyses, the mean scores of the scales will be used to indicate how participants perceive 1) preregistration (attitude scale), 2) subjective norms regarding preregistration (subjective norm scale), 3) their own control about using preregistration or not (behavioral control scale), and 5) their motivations (motivation scale) and 6) obstacles to preregister (obstacle scale).” (p. 16, l. 325 ff). Furthermore, when describing the analyses, we also added a short description of what the scales mean (cf. p. 23, l. 465 ff; p. 24, l. 492 ff).

Rev 3: I'd be interested to know how people respond to the question “A preregistration badge increases my trust in a study”. ...but this is becaues the question is related to my own research. Just thought I'd share the idea.

We added the item “A preregistration badge (i.e., a public acknowledgment that a study was preregistered provided by many journals) increases my trust in a study.” to the attitude scale.

 

Reviewer #4

Minor points

Rev 4: I think you undersell the role of other fields in open science. Preregistration, preprints and data sharing have a longer history and larger uptake in other fields. This doesn't diminish the importance of preregistration in psychology but it does suggest that psychology is not the only driving force for open science. 

We revised the introduction and emphasized the role of psychology for open science less.

Rev 4: You use brackets frequently and I sometimes found it quite difficult to read. May be worth moving some of the bracketed content into a full sentence to make it more readable.

We deleted many of the brackets and instead wrote the text in full sentences.

Rev 4: I suggest adding a “prefer not to answer” option to the survey's gender question. This will allow respondents protect their identity and may avoid some people dropping out.

A “prefer not to answer” option was included for the gender item.

Rev 4: I think your introduction is too long and duplicates existing review papers on the open science movement. Signposting readers to an appropriate open science review article could help cut your introduction down.

Thank you for pointing this out. We shortened the introduction substantially and think that it significantly improved the readability of our manuscript.

Rev 4: You sometimes cite opinion articles as evidence. For example “Preregistration also prioritizes theorizing and methods over results as the main criterion for judging the quality of research [20]”. van't Veer and Giner-Sorolla may have this priority but I do not know how many people who pre-register do. You also say “Analytical flexibility is decreased by defining and documenting planned analyses beforehand because this way one can clearly distinguish confirmatory from exploratory analyses [9,11,20,29].” You cite four opinion articles so again this is a belief of the authors not a fact. For statements like these, I would cite an empirical article supporting your statement (e.g. an article comparing outcome switching in pre-registered and unregistered clinical trials), remove it or qualify it.

Thank you for pointing out this important issue. We now make it more clear when we refer to opinions rather than empirical results. We removed some sources (as part of shortening the introduction) and included empirical evidence where possible.

Rev 4: I am unsure how you are using the terms “systematic evaluation focusing on preregistration” and “QRPs”. Definitions would help.

We removed the term "systematic evaluation" and instead described the aims and procedures in more detail (cf. p. 6, l. 101 ff). Additionally, we added a definition of "questionable research practices" (cf. p. 3, l. 45 ff) and refer the interested reader to other articles for more details.

Major points

Rev 4: I am nervous about how you will detect and control for possible selection bias from advertising the study on social media. Will you be collecting the recruitment method for each participant? If yes, will you examine and control for any differences between recruitment sources? How?

We will collect data about the recruitment method by distributing different links to researchers that were sourced from the different platforms. We plan to conduct some exploratory analyses using this data (e.g., reanalyze the data per platform).

Rev 4: The development and validity of your measures is unclear. You do not give existing validity evidence for measures you adapted from other survey. For the measures you created yourself, the validity evidence is unclear as is your rationale for the response options for these measures (for example why did you use 7-point scales for some and 4-point scales for others). To me, a more transparent description of your measures would allow others to replicate and evaluate them. Flake and Fried (2019, January 17) provide a list of questions to promote transparency in measurement that may be useful (https://psyarxiv.com/hs7wm/).

We now answer the questions suggested in the preprint in the manuscript (cf. p. 15 f, l. 305 ff). We also changed the scales so that all scale items are measured via a 7-point scale (before we decided to use a 4-point scale to resemble the work by Abele-Brehm et al., but there are many good reasons for using a 7-point scale as described in the manuscript). Thank you for this highly valuable comment, the preprint covers a very important topic.

Rev 4: I saw no reference to any piloting of the survey. If you have not piloted I recommend you do to understand, for example, when and why people drop out, if the survey is appropriate for people across different countries or non-native English speakers, and if people interpret the questions in the way you expect them to. 

We have conducted a pilot study and described all results in a new section pilot study (p. 26 f) in the manuscript. In the pilot study, we collected data from persons of various countries (complete datasets: n = 2 from Australia, n = 6 from America, n = 12 from Europe), none indicated problems understanding English. Additionally, items for which understanding seemed a bit ambiguous were revised. No specific patterns in dropout behavior were found (e.g., dropping out at a certain position).

Rev 4: How will you use data from the question “How would you define the term “preregistration”?”. You do not seem to plan to screen out people who understand preregistration differently to the open science definition but I recommend you do this. For example, you could describing preregistration at the start and include a “knowledge check” to ensure respondents are understanding it correctly.

We changed the study’s procedure: The item asking for a definition was deleted, and the new procedure now includes a given, precise definition and a knowledge check which, if not passed, shows the definition again in more depth (cf. p. 17, l. 340 ff).

Rev 4: Please be specific about what you powered your study to detect in the main body of the article. On line 200 you do not say what effect or effect size you have 95% statistical power to detect.

[L 199 ff: “The power analysis was specified to achieve a statistical power of 95% at a given significance threshold of 5% and a type II error probability of also 5% (α = β = .05).”]

We revised the power analysis section to be more precise about which N estimate is used as the overall targeted sample size, and additionally included a new figure (see Fig 1) that displays the individual analyses, the effect sizes for which they are powered, and the calculated sample sizes.

Rev 4: Your qualitative analyses are not described in enough detail for me to replicate them. To ensure others can replicate your analyses you can describe how you will analyse words quantitatively

We added a more detailed description of the qualitative analyses (cf. p. 20 f, l. 412 ff).

---

## [Decision Letter · Decision Letter 1]

11 Mar 2021

PONE-D-20-20445R1

Registered Report Protocol: Survey on attitudes and experiences regarding preregistration in psychological research

PLOS ONE

Dear Dr. Spitzer,

Thank you for submitting your manuscript to PLOS ONE. After careful consideration, we feel that it has merit but does not fully meet PLOS ONE’s publication criteria as it currently stands. Therefore, we invite you to submit a revised version of the manuscript that addresses the points raised during the review process.

**I would like to thank you for your revisions and to thank all 4 reviewers. **

**There are still some major points that need attention before we can move this registered report to next stage. **

We look forward to receiving your revised manuscript.

Kind regards,

Florian Naudet, M.D., M.P.H., Ph.D.

Academic Editor

PLOS ONE

Reviewers' comments:

Reviewer's Responses to Questions

**Comments to the Author**

1. Does the manuscript provide a valid rationale for the proposed study, with clearly identified and justified research questions?

Reviewer #1: Yes

Reviewer #2: Yes

Reviewer #3: Partly

Reviewer #4: Partly

2. Is the protocol technically sound and planned in a manner that will lead to a meaningful outcome and allow testing the stated hypotheses?

Reviewer #1: Yes

Reviewer #2: Yes

Reviewer #3: Partly

Reviewer #4: Partly

3. Is the methodology feasible and described in sufficient detail to allow the work to be replicable?

Reviewer #1: Yes

Reviewer #2: Yes

Reviewer #3: Yes

Reviewer #4: Yes

4. Have the authors described where all data underlying the findings will be made available when the study is complete?

Reviewer #1: Yes

Reviewer #2: Yes

Reviewer #3: Yes

Reviewer #4: Yes

5. Is the manuscript presented in an intelligible fashion and written in standard English?

Reviewer #1: Yes

Reviewer #2: Yes

Reviewer #3: Yes

Reviewer #4: Yes

6. Review Comments to the Author

You may also provide optional suggestions and comments to authors that they might find helpful in planning their study.

Reviewer #1: COMMENTS: Since most of the comments made on earlier draft by me (and hopefully by other respected reviewers also) are attended positively/adequately, I am fully satisfied and the manuscript is improved a lot.

Particularly, what you said (while answering my comment) [When the Stage 2 Registered Report is submitted, the sections “results” and “discussion” will be added and we may move confirmed hypotheses to the results section] is perfectly right/ appropriate. Change made in ‘title’ [“Registered Report Protocol: Survey on ……] is highly appreciated. Changes made in the “scales” also are appreciable/very good.

Reviewer #2: I thank the authors for addressing all points. I know how much time it has taken, especially in the case of a registered report this can be agonizing. However, this project is a nice piece of research and I am looking forward to the results.

Reviewer #3: The revised Stage 1 RR manuscript has been improved in many ways. I applaud the authors for their thorough work. There remain, however, at least two major concerns.

1. Sampling bias. The authors state that they are not concerned about sampling bias:

“this is not an issue for our survey because the main focus does not lie on gaining a representative sample of all psychological researchers, but on identifying problems in the current implementation of preregistration and uncovering reasons for psychological researchers’ potential ambivalence toward it.”

Based on their sampling plan, the sample will be heavily biased towards researchers who have already preregistered and are familiar with it. 50% of their sample will come from OSF registries, and thus by definition have already preregistered, and 90% of their pilot sample have experience with preregistration. Thus, their sentence above would need to be modified to state: “…uncovering reasons for psychological researchers’ [who have already used preregistration] potential ambivalence toward it.” I find that clearly stating this limited generalizability due to the selective/biased sample makes the question seem much less interesting. Similarly, the hypotheses become much less interesting (e.g., “1.1. [Of researchers who are familiar with preregistration,] More beneficial attitudes are a positive predictor for the intention to preregister”. “2.2. Early career researchers (PhD students) [who have used preregistration] differ from senior researchers (postdocs, professors) [who have used preregistration] in their attitudes (i.e., how positive they view preregistration).”

All this to say, the study could be run with a sample of participants that are familiar with preregistration, but all the conclusion would need to be tempered to ensure that they don’t make claims beyond the selective sample. I know PLOS ONE focuses on ensuring that the methods are sounds and is largely agnostic to the importance of research, but I am concerned that this sampling bias would inhibit the researcher from answering the question they are interested in answering. I recommend aiming for a representative sample of psychology researchers. This could be done by using Web of Science, Scopus, and other databases (as a side note, PubMed may lead to a sample of psychology researcher more concerned with the medical side of psychology).

If the authors decide to continue to sample from the OSF REGISTRIES, I strongly believe that this sample should be analyzed separately from those from the Web of Science and PubMed samples, because they are drawn from what I would consider different populations. Thus, summary statistics that combine these samples don’t represent a real population.

2. Sampling plan and pilot study. The authors calculate their sampling plan with a 10% response rate from the pilot study. However, there were no Master’s students or postdocs who completed the survey, and thus the response rate is 0% for these groups. The authors do not report how many participants in the pilot were from the 3 samples of OSF, Web of Science, and PubMed. For their sampling plan to work as hoped, the response rate would need to be 10% from each of these platforms in each of the 4 career stages, which their data show is not the case. Furthermore, it is unclear how additional participants from social media and mailing lists will be used in the analysis. These additional participants introduce an additional sampling bias.

Minor comments:

1. Line 64. Preregistration is mandated in medicine, although I’m not sure if it’s the norm. Many studies are still not registered or they are registered retrospectively.

2. Explain how scores such as “beneficial attitudes” will be calculated. Will it just be the mean of all those survey questions?

3. Your specific usage of the term Early Career Research will confuse readers as this term almost always encompasses postdocs. I recommend simply saying PhD students.

4. Will your study include research associates, readers and lecturers (i.e., non-full professors)?

On another note, I think that the survey responses presented as descriptive statistics would be interesting even if the sample was biased.

In summary, I believe (1) the sampling bias needs to be minimized, or the authors should present a strong argument for why sampling bias is not a concern (which I don’t believe has been presented so far). And (2) that evidence is needed that the sampling plan will be sufficient to achieve the desired numbers in each category; perhaps demonstrable with a larger pilot and revised categories or sample size.

I am sympathetic to research on preregistration and also conduct some with my group. I hope these comments are taken in the collegial manner intended.

I always sign my reviews,

Robert Thibault

Reviewer #4: I responded "Partly" to questions 1 and 2 because the validity of the measures in the survey remain a key concern for me. I would also like to know more about the theoretical basis of the study. See the attached comments document for more information.

7. PLOS authors have the option to publish the peer review history of their article (what does this mean?). If published, this will include your full peer review and any attached files.

Reviewer #1: **Yes: **Dr. Sanjeev Sarmukaddam

Reviewer #2: No

Reviewer #3: **Yes: **Robert T. Thibault

Reviewer #4: **Yes: **Katie Drax

---

## [Author Response · Author response to Decision Letter 1]

23 Apr 2021

Response to reviewers

We thank the reviewers for reading our revised manuscript thoroughly and for providing constructive criticism. We believe that by addressing all points to our best knowledge, we were able to improve the study further, and we hope that you now feel that the revised manuscript can be recommended for an in-principle-acceptance. 

Overall, we have implemented various strategies to counter a potential sampling bias by adjusting both our recruitment strategy and our analyses. We further updated our exclusion criteria and added a sensitivity analysis. Furthermore, we decided to change our approach toward the distinction of early career vs. senior researchers by rather using the more continuous variable of research experience, i.e., the number of years someone has engaged in psychological research. Accordingly, we will compute regression models investigating attitudes and the perceived strength of motivations and obstacles (hypothesis 2) rather than a MANOVA. Since research experience is measured by a new item that was not included in the previous version of the survey, we will not use the pilot data for the main study.

More details about the changes are included in our responses to each reviewers’ comments below. If line numbers were given by the reviewers, we include the term, sentence, or section, to which the comment refers to, in parentheses. 

Reviewer #1

Rev 1: Since most of the comments made on earlier draft by me (and hopefully by other respected reviewers also) are attended positively/adequately, I am fully satisfied and the manuscript is improved a lot.

Particularly, what you said (while answering my comment) [When the Stage 2 Registered Report is submitted, the sections “results” and “discussion” will be added and we may move confirmed hypotheses to the results section] is perfectly right/ appropriate. Change made in ‘title’ [“Registered Report Protocol: Survey on ……] is highly appreciated. Changes made in the “scales” also are appreciable/very good.

We are very pleased that you feel we have improved the manuscript and that you are now fully satisfied with it. Thank you very much for your constructive comments, which helped us a lot in the revision process.

Reviewer #2

Rev 2: I thank the authors for addressing all points. I know how much time it has taken, especially in the case of a registered report this can be agonizing. However, this project is a nice piece of research and I am looking forward to the results.

We would like to thank you for your valuable assistance with our study. We are very pleased that you find our research interesting and we are also looking forward to the results.

Reviewer #3

Rev 3: The revised Stage 1 RR manuscript has been improved in many ways. I applaud the authors for their thorough work. There remain, however, at least two major concerns.

1. Sampling bias. The authors state that they are not concerned about sampling bias:

“this is not an issue for our survey because the main focus does not lie on gaining a representative sample of all psychological researchers, but on identifying problems in the current implementation of preregistration and uncovering reasons for psychological researchers’ potential ambivalence toward it.”

Based on their sampling plan, the sample will be heavily biased towards researchers who have already preregistered and are familiar with it. 50% of their sample will come from OSF registries, and thus by definition have already preregistered, and 90% of their pilot sample have experience with preregistration. Thus, their sentence above would need to be modified to state: “…uncovering reasons for psychological researchers’ [who have already used preregistration] potential ambivalence toward it.” I find that clearly stating this limited generalizability due to the selective/biased sample makes the question seem much less interesting. Similarly, the hypotheses become much less interesting (e.g., “1.1. [Of researchers who are familiar with preregistration,] More beneficial attitudes are a positive predictor for the intention to preregister”. “2.2. Early career researchers (PhD students) [who have used preregistration] differ from senior researchers (postdocs, professors) [who have used preregistration] in their attitudes (i.e., how positive they view preregistration).”

All this to say, the study could be run with a sample of participants that are familiar with preregistration, but all the conclusion would need to be tempered to ensure that they don’t make claims beyond the selective sample. I know PLOS ONE focuses on ensuring that the methods are sounds and is largely agnostic to the importance of research, but I am concerned that this sampling bias would inhibit the researcher from answering the question they are interested in answering. I recommend aiming for a representative sample of psychology researchers. This could be done by using Web of Science, Scopus, and other databases (as a side note, PubMed may lead to a sample of psychology researcher more concerned with the medical side of psychology).

If the authors decide to continue to sample from the OSF REGISTRIES, I strongly believe that this sample should be analyzed separately from those from the Web of Science and PubMed samples, because they are drawn from what I would consider different populations. Thus, summary statistics that combine these samples don’t represent a real population.

Thank you for your thorough feedback. We have thought deeply about this criticism and think that it is very helpful that you addressed and highlighted it. To counter this bias, we will use the following strategies:

We will increase recruiting through general databases to achieve a sample that resembles the real population of psychological researchers better. We consulted one of our in-house experts from the department of information services of our institute for this purpose and decided to use the following databases: Web of Science, PubMed, PSYNDEX, and PsycInfo. Through each of these databases, 22.5% of participants will be invited (90% in total). Meanwhile, the percentage invited through OSF will be significantly reduced to 10% instead of 50%. Of course, recruitment via OSF could be eliminated altogether, however, Hardwicke et al. (2020) have shown that it might be the case that only very few studies are preregistered. So it could be that in the general sample hardly any researchers preregistered. Since we are also interested in the experiences of researchers who prepared and submitted a preregistration themselves, we decided to still invite a small fraction of the sample via OSF (10%) to ensure a subsample with preregistration experience. We have described this in detail in the section data collection and stopping rule (p. 12 ff).

Nevertheless, this recruitment strategy might still lead to a bias. We will control for this in the following way: We will make the general statement of what proportion of researchers have already preregistered only for the sample recruited via the general databases (i.e., we achieve a representative statement here, as we exclude the “biased” OSF sample). We will run all further descriptive reports separately for researchers who have preregistered before vs. not and compare between these groups (see section descriptive reports, p. 20 ff). We also want to account for possible biases in our hypotheses tests and therefore decided to include “preregistration experience” as a dichotomous control variable (see section hypotheses tests, p. 22 ff). 

By including these various control strategies, we think that we now control well for the potential sampling bias.

Rev 3: 2. Sampling plan and pilot study. The authors calculate their sampling plan with a 10% response rate from the pilot study. However, there were no Master’s students or postdocs who completed the survey, and thus the response rate is 0% for these groups. The authors do not report how many participants in the pilot were from the 3 samples of OSF, Web of Science, and PubMed. For their sampling plan to work as hoped, the response rate would need to be 10% from each of these platforms in each of the 4 career stages, which their data show is not the case. Furthermore, it is unclear how additional participants from social media and mailing lists will be used in the analysis. These additional participants introduce an additional sampling bias.

We conducted the pilot study because we wanted to know more about the overall response rate and test the recruitment method and the survey items. We don’t want to draw conclusions about more fine-grained distributions such as the return rate for different groups or databases based on this small sample (we also deleted this information from the manuscript as it might be misleading). However, our recruitment strategy anticipates that different academic groups will not be equally represented. To compensate this, we set quota variables for different academic degrees and intend to explicitly recruit participants for the open quotas in the second wave of invitations (see Fig 2). 

Regarding the databases, we do not employ quotas as we believe that participants whose publications appear on the general databases (Web of Science, PubMed, PSYNDEX, PsycInfo) will overlap (though duplicates will be excluded) and we have no reason to believe that their attitudes toward preregistration or their response rate differ depending on the database. This is different for participants recruited from OSF Registries. Here we suppose that attitudes toward preregistration will be more positive as these participants were identified based on their preregistration. To counter this bias, we will include preregistration experience in the respective analyses as outlined above. This may also help, at least partly, in terms of a self-selection bias, that is, that people who are inherently more interested in preregistration may be more likely to participate in the survey. This could also apply to people who were recruited from social media. Additionally, we will distribute different survey links per database and for social media posts which will allow us to infer the recruitment source and thus, to check post-hoc for differences. 

Minor comments:

Rev 3: 1. Line 64. Preregistration is mandated in medicine, although I’m not sure if it’s the norm. Many studies are still not registered or they are registered retrospectively.

[L 64 ff: However, while preregistration is already the norm in other scientific disciplines like medicine [18], it has been frequently demanded as a means to counter questionable research practices but is still not widely practiced in psychology.]

We have changed the sentence to “However, while preregistration is already well-established in other scientific disciplines and is mandated, for example, in medicine [18], it has been frequently demanded as a means to counter questionable research practices but is still not widely practiced in psychology.” (p. 4, l. 64 ff).

Rev 3: 2. Explain how scores such as “beneficial attitudes” will be calculated. Will it just be the mean of all those survey questions?

Yes, we will calculate the attitude score by computing the mean for each participant over all attitude items (24 items). The same procedure will be done for the subjective norm scale (mean of eight subjective norm items), the perceived behavioral control scale (mean of five perceived behavioral control items), the intention scale (mean of three intention items), the motivation scale (mean of ten motivation items), and the obstacle scale (mean of ten obstacle items). We made this clearer in the manuscript in the section material (p. 15 ff): “For the statistical analyses, the mean scores of the scales will be used to measure how participants perceive 1) preregistration (attitude scale), 2) subjective norms regarding preregistration (subjective norm scale), 3) their own control about using preregistration or not (behavioral control scale), 4) their intention to use preregistration in the future (intention scale), and 5) their motivations (motivation scale) and 6) obstacles to preregister (obstacle scale). For each participant, the mean for each scale will be calculated and used as the score.” (p. 17, l. 338 ff).

Rev 3: 3. Your specific usage of the term Early Career Research will confuse readers as this term almost always encompasses postdocs. I recommend simply saying PhD students.

We are very thankful for this comment, and have thought about it in depth. We have concluded that our classification into academic groups was ambiguous and the distinction between early career researches and senior researchers unnatural since experience may be better represented on a more continuous scale. Therefore, we decided to change the operationalization of research experience into a more objective one: We will use the highest academic degree in psychology for the descriptive reports, and define research experience by the years someone has already worked in psychological research. 

Hypothesis 2 will consequently be tested via multiple regressions with research experience as a predictor for 1) attitudes, 2) motivations, and 3) perceived intensity of obstacles (see section hypotheses tests, p. 22 ff). These will replace the previously planned MANOVA.

The preregistration experience will be included as a control variable. We hope to achieve more meaningful conclusions by using a more objective, clear, cross-regionally transferable, and continuous variable as predictor for our scales.

Rev 3: 4. Will your study include research associates, readers and lecturers (i.e., non-full professors)?

We have decided against using our previous classification altogether (see our comment above). For the descriptive reports, we will group participants based on their degree (bachelor's degree, master's degree, doctoral degree, habilitation or full professorship), and for our hypotheses tests, we will use research experience defined as the years someone has already worked in psychological research.

Rev 3: On another note, I think that the survey responses presented as descriptive statistics would be interesting even if the sample was biased.

In summary, I believe (1) the sampling bias needs to be minimized, or the authors should present a strong argument for why sampling bias is not a concern (which I don’t believe has been presented so far). And (2) that evidence is needed that the sampling plan will be sufficient to achieve the desired numbers in each category; perhaps demonstrable with a larger pilot and revised categories or sample size.

I am sympathetic to research on preregistration and also conduct some with my group. I hope these comments are taken in the collegial manner intended.

I always sign my reviews,

Robert Thibault

Once again, we would like to thank you for the extensive work you have put into your reviews of our study so far. We have thought thoroughly about your comments and tried to address them to our best knowledge. We hope that you now find our recruitment strategy and categories more appropriate, and are looking forward to your feedback. 

Reviewer #4

Rev 4: I responded “Partly” to questions 1 and 2 because the validity of the measures in the survey remain a key concern for me. I would also like to know more about the theoretical basis of the study. See the attached comments document for more information.

It was a pleasure reviewing this again. You have done an admirable job responding to my original comments and I’m glad that the pilot study was useful for you. Overall, I think your responses are excellent in that they have provided more clarity and justification for your proposed method.

Below I have outlined my thoughts on points I brought up in the previous review cycle and new issues.

Thank you very much for your previous comments and for your feedback. We have tried to address your comments in detail, as described below.

Previous comments

Rev 4: You have resolved all my previous minor points and my previous major points about piloting, the power analyses, and qualitative analysis.

We are happy that you are satisfied with our implementation of your previous comments.

Rev 4: Lines 244-249 imply to me that “a sample biased toward researchers who have already preregistered… is not an issue.. on identifying problems in the current implementation of preregistration and uncovering reasons for psychological researchers’ potential ambivalence toward it.”. I agree a biased sample may help identify some problems and reasons but only the ones relevant to the sample. We will not know if or how they exist in the wider population. A major issue in the sample may only be minor in wider population. Reframing these lines as a limitation would be clearer.

[L 244 ff: This recruitment procedure might result in a sample biased toward researchers who have already preregistered (see section pilot study). However, this is not an issue for our survey because the main focus does not lie on gaining a representative sample of all psychological researchers, but on identifying problems in the current implementation of preregistration and uncovering reasons for psychological researchers’ potential ambivalence toward it.]

We agree and since we indeed want to, at least partly, make statements about the wider population of psychological researchers, we decided to more deliberately protect against this sampling bias. In particular, we amended our recruitment strategy by including more general databases and by inviting less participants via OSF Registries. To examine the percentage of preregistrations in the wider population, the subsample from OSF Registries will be excluded or reported separately. When testing our planned hypotheses, we will include preregistration experience as a control variable (please see our detailed response to a similar comment by reviewer #3 above). By using these new approaches, we hope to be able to make statements about a larger part of the psychological research community. Persisting limitations will be addressed when discussing the results.

Rev 4: I do not understand the rationale for the numbers of items or the process by which you came to that number (24 items for attitudes, 1 for perceived importance, 8 for subjective norm, etc). Making this clearer will help readers understand the justification for the measures and the survey length. 

For our scales, we took a content-guided approach. Therefore, the number of items is determined by how many different aspects a scale covers. Since we assume, for example, that attitudes are a very complex construct, we have included a larger number of items for this scale in order to be able to represent this complexity. Meanwhile, the motivation and obstacle scale have the same length (ten items) to evenly display motivations and obstacles. Lastly, the number of items of the subjective norm scale (eight items) and the perceived behavioral control scale (five items) was chosen to reflect some of the most important aspects of these constructs, without targeting a specific number of items.

Following your comment, we have studied a manual for constructing TPB questionnaires to ensure that our design is appropriate. Based on the manual (“Constructing questionnaires based on the theory of planned behaviour: A manual for health services researchers”, Francis et al., 2004), we have added two questions inquiring about the intention to use preregistration in the future. We will now use the mean score of this intention scale as dependent variable for hypothesis 1 (rather than the response to the single item that we had before). Our other items were already in accordance with the guide. We hope that using this guide further improves the validity of our questionnaire.

Rev 4: It is unclear to me how you chose the options to the non-scale items (e.g. drawbacks of preregistration). Providing a rationale for them would allow readers to replicate and evaluate the measures. The options could also be quite leading. Changing all such items to open text and analysing them qualitatively would reduce the risk of options being leading.

Our initial idea was to reduce the effort for the participants and obtain a ranked order of the options that were selected based on literature research and personal experience. However, we agree that providing options may limit the responses of participants and thus decided to retract to open text inputs as suggested.

Specifically, given answer options were deleted for the items: 

1) What are positive/negative consequences of mandatory preregistration? 

2) Benefits of preregistration

3) Drawbacks of preregistration

4) Reasons against preregistration

Nevertheless, we would like to keep multiple choices for the two items addressing problems and worries in the context of preregistration (i.e., the questions “Did you encounter specific problems when preregistering a study? If yes, which ones?” and “What worries do you have with respect to preregistering your studies?”) as they are matched. That is, the same aspects are mentioned in both items but the items are presented to different groups: Researchers who have not yet preregistered are asked about their worries and researchers who have preregistered their studies in the past are asked about the problems that they encountered. This matching makes it possible to specifically compare worries and experienced problems.

New comments – minor points

Rev 4: Line 52 - I would change this to “Preregistration – on the rise?” 

[L 52: Preregistration – At the rise in psychology?]

We have updated the section title accordingly (cf. p. 4, l. 52).

Rev 4: I didn’t see an explanation of how you will identify and exclude “bot” responses. Including a statement about this would make this clear – see this tweet for further details https://twitter.com/m_simonephd/status/1174010078632009728

We included a captcha question at the start of the survey (i.e., an arithmetic task, cf. p. 18, l. 352 f; we decided against using the recaptcha plugin by Google due to privacy concerns). Furthermore, we will distribute different links to researchers recruited via the different databases (i.e., OSF Registries, Web of Science, PubMed, PSYNDEX, PsycInfo) and social media.

Rev 4: I would probably fail the “knowledge check” because I would not check the box that “preregistration will increase transparency”. I believe preregistration could increase transparency, but that this is not a guarantee. “Bad” and poorly detailed preregistrations are possible. Removing the “increases transparency” check from the knowledge check question would avoid people like me failing the check.

We changed the option to “Preregistration aims at increasing the transparency of potential changes made to a study.” to reduce ambiguity.

Rev 4: If I was a participant, I would find it helpful to have labels above each of the 7-point items (e.g. strongly disagree – disagree- somewhat disagree – neither agree nor disagree – somewhat agree – agree – strongly agree). Otherwise, I would not be sure what the “neutral” answer was, or I’d have to count along the row to find the neutral answer.

For all scale items, the suggested labels were included (however, “slightly” was used instead of “somewhat”). The labels included now are: “strongly disagree - disagree - slightly disagree - neither agree nor disagree - slightly agree - agree - strongly agree” (also see S4 and S5 Videos).

Rev 4: Sensitivity analysis to examine how your exclusions affect the results could be useful

All participants that indicate that their research or studies do not fall within the scope of psychology or that do not have at least a bachelor’s degree in psychology, and thus, cannot be assigned to the respective quota, will be screened out at the beginning of the survey, thus, in this case, no sensitivity analysis is possible. However, for our other exclusion criteria, we will conduct sensitivity tests. Specifically, we will repeat the hypotheses tests while also including participants that have not completed all pages of the survey (see section exclusion and missing data, p. 14 f).

Rev 4: Lines 277-281. It is unclear to me how excluding masters students who do not intend to pursue an academic career will mean that your survey will target a “research-oriented psychology” population. What about members of the other groups who intend to leave academia very soon (especially relevant for PhD students)? The question also means you run the risk of excluding master’s students who don’t intend to go into academia but ultimately do, and including those who do intend to continue but don’t. I recommend you clarify why you are not excluding other groups who intend to leave academia soon and how you will handle the risk of master’s students’ intentions not matching their behaviour. Alternatively, you could ask members of all groups if they intend to pursue an academic career and conduct sensitivity analyses to see if differences occur based on this intention.

[L 275 ff: All participants that indicate 1) that their research does not fall within the scope of psychology (or for master students, whose major is not psychology), 2) that they are neither a master student, PhD student, postdoc nor professor, or 3) that they are master students who do not intend pursuing an academic career, will be screened out at the beginning of the survey. Thus, they will be directed to an exit page rather than to the main body of the survey, will not be counted into the quotas, and will not be considered for data analyses as the survey targets a sample from research-oriented psychology.]

As suggested, we will include all psychologists with at least a bachelor’s degree (formerly the group of master students) and present the question: “Do you plan to continue your academic career, or leave academia? If you are unsure, please indicate your current tendency here.” to all participants for possible post-hoc analyses.

New comments – major points

Rev 4: The introduction contains little discussion of relevant theories and their respective strengths and limitations. I think there has been lively debate around the utility and validity of the theory of planned behaviour (e.g. https://www.tandfonline.com/doi/full/10.1080/17437199.2013.869710). I would understand the theoretical basis of your survey better if you explained why you chose the theory of planned behaviour over others, if the debates about the theory of planned behaviour apply and how you will handle them.

Thank you for pointing this out. We decided to use the TPB as the basis for our first hypothesis since it is still widespread, relevant, and promising today. For example, it has been argued to be the most widely used theory in the field of human decision research (e.g., see “The Enduring Use of the Theory of Planned Behavior”, Miller, 2017). Supporting this claim, a large pool of studies has addressed the theory of planned behavior, as demonstrated by a search on Web of Science showing that 2029 articles which contain the keywords “theory of planned behavior” have been published in the last five years alone. Furthermore, it has been tested numerous times by meta-analytical approaches with promising results (e.g., see [40–45] of our manuscript). 

We will acknowledge potential shortcomings of the theory in the following way: While its flexibility is often pointed out as a virtue of the theory, the parsimony of its variables is also criticized. We will extend the theory by adding a moderating variable (“perceived importance of preregistration”) as well as a control variable (“preregistration experience”) to make better predictions. Another criticism is that the theory is not falsifiable, especially since in the case of null effects researchers oftentimes criticize their study design rather than the theory. Our study is a Registered Report and is reviewed and revised for methodological rigor before it is conducted. We will take this into consideration and be careful to not interpret null effects as design flaws only, but such that the theory might not be able to predict intentions in our context. Other concerns were voiced regarding validity and utility of the model. In this regard, we refer to the meta-analyses pointed out in the manuscript (cf. p. 7, l. 127) which we believe highlight the benefits of the theory.

We have added a summary of our arguments for using the theory in the manuscript (cf. p. 6, l. 115 ff).

Rev 4: Measurement validity remains a key concern for me but for more reasons than before. I’m glad you found the Flake & Fried paper useful and lines 317-320 now make the validity of the survey clear. The lack of validity measures mean I feel I do not have evidence that your measures are measuring what you intend them to. I believe this will weaken the evidence your study can provide but I understand this may be an unavoidable limitation. 

Thank you for pointing this out. We fully agree with your concern, and we will mention this limitation in the discussion section of our Registered Report Article.

Rev 4: I understand that me bring up new issues on the second round may be difficult but I hope they are still useful. 

Kate Drax

Thank you for both your feedback to your previous comments and your new comments as we feel that your suggestions further improved our planned study. We are now looking forward to your feedback.

---

## [Decision Letter · Decision Letter 2]

20 May 2021

PONE-D-20-20445R2

Registered Report Protocol: Survey on attitudes and experiences regarding preregistration in psychological research

PLOS ONE

Dear Dr. Spitzer,

Thank you for submitting your manuscript to PLOS ONE. After careful consideration, we feel that it has merit but does not fully meet PLOS ONE’s publication criteria as it currently stands. Therefore, we invite you to submit a revised version of the manuscript that addresses the points raised during the review process.

We look forward to receiving your revised manuscript.

Kind regards,

Florian Naudet, M.D., M.P.H., Ph.D.

Academic Editor

PLOS ONE

Journal Requirements:

Reviewers' comments:

Reviewer's Responses to Questions

**Comments to the Author**

1. Does the manuscript provide a valid rationale for the proposed study, with clearly identified and justified research questions?

Reviewer #3: Yes

Reviewer #4: Yes

2. Is the protocol technically sound and planned in a manner that will lead to a meaningful outcome and allow testing the stated hypotheses?

Reviewer #3: Yes

Reviewer #4: Partly

3. Is the methodology feasible and described in sufficient detail to allow the work to be replicable?

Reviewer #3: Yes

Reviewer #4: Yes

4. Have the authors described where all data underlying the findings will be made available when the study is complete?

Reviewer #3: Yes

Reviewer #4: Yes

5. Is the manuscript presented in an intelligible fashion and written in standard English?

Reviewer #3: Yes

Reviewer #4: Yes

6. Review Comments to the Author

You may also provide optional suggestions and comments to authors that they might find helpful in planning their study.

Reviewer #3: The authors addressed my main concern about sampling bias. My concern about achieving their sample size remains. Based on their pilot data, and my best guess, they will need to invite many more than the planned 2960 authors if they want to have 25% of 296 participants in each category (Bachelors, Master’s, doctoral, professorship).

For all hypotheses 1.1-1.6, it could be interesting to see if the variable “have used preregistration in the past” could account for the effects you might find. We can’t establish causality with these hypotheses (i.e., whether the intention to register causes the beneficial attitudes or vice versa). If you find an effect, but you also find that this effect is eliminated when accounting for having used preregistration in the past, than another potential explanation of the results could simply be “those who registered in the past have more beneficial attitudes towards preregistration AND a greater intention to preregister again”.

Other comments.

• I’m not familiar with the term “habilitation”.

• [line 275] Will “psychology” be used as a term in the search query box and if so within which fields (e.g., Title, keywords, abstract), or will it be used as a filter that the websites allow you to select from.

• I suggest you keep the phrase you deleted, it is relevant information. “(17 PhD students, three postdocs, seven professors, and two members of other academic groups which were screened out)

• The new question in response to Reviewer 4 still does not solve the issue they raise: [We changed the option to “Preregistration aims at increasing the transparency of potential changes made to a study.” to reduce ambiguity.] Some people may disagree or see this as a less important purpose of preregistration. For example, other reasons are to share what research is being done to reduce duplication or to help develop solid analysis plans.

I always sign my reviews,

Robert Thibault

Reviewer #4: The authors have done an impressive job at responding to all author comments and I thank them for responding to all of my comments so completely. I still have reservations about the theoretical basis of the study and the validity of the measures but, thanks to the authors responses, I look forward to the results of the study.

7. PLOS authors have the option to publish the peer review history of their article (what does this mean?). If published, this will include your full peer review and any attached files.

Reviewer #3: **Yes: **Robert T. Thibault

Reviewer #4: **Yes: **Kate Drax

---

## [Author Response · Author response to Decision Letter 2]

7 Jun 2021

Response to reviewers

We thank the reviewers for reading our revised manuscript thoroughly and for providing constructive comments and criticism. We believe that by addressing all points to our best knowledge, we were able to improve the study further, and we hope that you now feel that the revised manuscript can be recommended for an in-principle-acceptance. 

Overall, we have further improved our recruitment strategy to ensure that we will reach our targeted sample size, made small changes to the survey to improve its comprehensibility, and added some clarifying descriptions to our manuscript. Furthermore, we reviewed our reference list.

More details about the changes are included in our responses to each reviewers’ comments below. If line numbers were given by the reviewers, we include the term, sentence, or section, to which the comment refers to, in parentheses.

Reviewer #3

R3: The authors addressed my main concern about sampling bias. My concern about achieving their sample size remains. Based on their pilot data, and my best guess, they will need to invite many more than the planned 2960 authors if they want to have 25% of 296 participants in each category (Bachelors, Master’s, doctoral, professorship).

We are very pleased that we could address your doubts regarding the sampling bias. Thank you for voicing your concern about our planned recruitment strategy. We amended our invitation strategy in the following way: In the second invitation wave, we will consider the actual response rate of the first invitation wave to determine how many more participants of open quotas need to be invited. This is described in the section data collection and stopping rule: “If this does not result in a sufficient sample size, an additional invitation wave will be conducted. In this second wave, participants will subsequently be invited for quotas that have not yet been filled. For each open quota, the response rate of the first invitation wave will be used to calculate how many more participants need to be invited to fill the quota.” (p. 13 f, l. 262 ff) and in more detail in the Supporting Information (see S2 Text): “To determine how many people need to be recruited in the second wave to fill the quotas, we will use the response rate from the first invitation wave. This can be illustrated by an example: Since we are aiming for 25% quotas, our final sample should contain n = 74 participants from each quota (25% of our targeted sample size N = 296). If after the first invitation wave, only 64 members of a quota have participated, this would mean that ten people are still missing in this quota. To calculate how many more participants need to be invited to fill the quota, the overall response rate of the first invitation wave will be used: If only 237 (instead of 296) of the 2960 invited people have participated, this would correspond to an actual response rate of 8% instead of the estimated 10%. If we assume that 8% of invited researchers will also respond in the second invitation wave, we would need to invite 125 more participants to fill our example quota (since 10 is 8% of 125). Calculations of this kind will be done for all open quotas after the first invitation wave, to determine how many more participants to invite. However, due to reasons of practicality, no more than 740 more participants (i.e., 25% of the first invitation wave) will be invited per quota in the second wave.”.

R3: For all hypotheses 1.1-1.6, it could be interesting to see if the variable “have used preregistration in the past” could account for the effects you might find. We can’t establish causality with these hypotheses (i.e., whether the intention to register causes the beneficial attitudes or vice versa). If you find an effect, but you also find that this effect is eliminated when accounting for having used preregistration in the past, than another potential explanation of the results could simply be “those who registered in the past have more beneficial attitudes towards preregistration AND a greater intention to preregister again”.

Thank you for your comment. In case the control variable “preregistration experience” is significant, we will conduct additional exploratory analyses to further investigate potential directions of effects. If these result in the conclusion “those who registered in the past have more beneficial attitudes toward preregistration AND a greater intention to preregister again,” we would be able to conclude that preregistering in the past does not discourage one from preregistering in the future (i.e., we can rule out that researchers have had negative experiences that would discourage them from preregistering again), which we think would also be a valuable information.

Other comments.

R3: I’m not familiar with the term “habilitation”.

Habilitation is a qualification required to obtain a professorship in many European countries. To allow individuals unfamiliar with habilitation to answer the question, "full professorship" is also included in the response: “Habilitation and/or full professorship”.

R3: [line 275] Will “psychology” be used as a term in the search query box and if so within which fields (e.g., Title, keywords, abstract), or will it be used as a filter that the websites allow you to select from.

[L 235: The term “psychology” will be searched for on specified databases, and resulting hits will be sorted from newest to oldest.]

The former. To provide all details while still keeping the description reasonably concise, we included this information in the Supporting Information (see S2 Text): “On Web of Science, PubMed, PSYNDEX, and PsycInfo, the search term will be set to search “all fields”, it will be searched for the keyword “psychology”, and documents will be set to articles, to get a broad image of research articles focusing on psychology. Documents will be sorted from new to old. All articles that do not focus on psychology will be excluded. On OSF Registries, it will also be searched for the keyword “psychology” and documents will be sorted from new to old.”

R3: I suggest you keep the phrase you deleted, it is relevant information. “(17 PhD students, three postdocs, seven professors, and two members of other academic groups which were screened out)

[L 550 f: In this time, 29 participants started the survey of which 20 completed it, yielding an overall response rate of 10%.]

We added this part of the sentence again in the manuscript: “In this time, 29 participants (17 PhD students, three postdocs, seven professors, and two members of other academic groups which were screened out) started the survey of which 20 completed it (14 PhD students and 6 professors), yielding an overall response rate of 10%.” (p. 27, l. 553 ff).

R3:The new question in response to Reviewer 4 still does not solve the issue they raise: [We changed the option to “Preregistration aims at increasing the transparency of potential changes made to a study.” to reduce ambiguity.] Some people may disagree or see this as a less important purpose of preregistration. For example, other reasons are to share what research is being done to reduce duplication or to help develop solid analysis plans.

We understand that there might be different approaches of how to define preregistrations, and what are important aspects of preregistration. This is precisely why we have chosen to present a definition in our survey, so that all participants know what aspects of preregistration we are referring to and to ensure consistent understanding. We have made it clear in the survey that the knowledge check specifically targets the understanding of our presented definition (“Which aspects belong to the definition of “preregistration”, based on the definition on the previous page? This question’s purpose is to check if you understood our definition of “preregistration” correctly. There are, of course, many ways to define preregistration. However, please base your answer on our definition and keep this definition in mind when answering the survey. Please select every correct option.”). However, even if participants answer this question incorrectly, they are not excluded but are only shown the definition again, and are then redirected to the main survey.

Reviewer #4

R4: The authors have done an impressive job at responding to all author comments and I thank them for responding to all of my comments so completely. I still have reservations about the theoretical basis of the study and the validity of the measures but, thanks to the authors responses, I look forward to the results of the study.

Thank you for your review, which has helped us immensely to improve our study. We agree that there are some limitations which we will fully disclose when discussing the results of our study.

---

## [Decision Letter · Decision Letter 3]

17 Jun 2021

Registered Report Protocol: Survey on attitudes and experiences regarding preregistration in psychological research

PONE-D-20-20445R3

Dear Dr. Spitzer,

We’re pleased to inform you that your manuscript has been judged scientifically suitable for publication and will be formally accepted for publication once it meets all outstanding technical requirements.

Kind regards,

Florian Naudet, M.D., M.P.H., Ph.D.

Academic Editor

PLOS ONE

Additional Editor Comments (optional):

Reviewers' comments:

Reviewer's Responses to Questions

**Comments to the Author**

1. Does the manuscript provide a valid rationale for the proposed study, with clearly identified and justified research questions?

Reviewer #3: Yes

2. Is the protocol technically sound and planned in a manner that will lead to a meaningful outcome and allow testing the stated hypotheses?

Reviewer #3: Yes

3. Is the methodology feasible and described in sufficient detail to allow the work to be replicable?

Reviewer #3: Yes

4. Have the authors described where all data underlying the findings will be made available when the study is complete?

Reviewer #3: Yes

5. Is the manuscript presented in an intelligible fashion and written in standard English?

Reviewer #3: Yes

6. Review Comments to the Author

You may also provide optional suggestions and comments to authors that they might find helpful in planning their study.

Reviewer #3: I have no further comments. Looking forward to reading the results!

7. PLOS authors have the option to publish the peer review history of their article (what does this mean?). If published, this will include your full peer review and any attached files.

Reviewer #3: **Yes: **Robert T. Thibault

---

## [Editor Report · Acceptance letter]

24 Jun 2021

PONE-D-20-20445R3 

Registered Report Protocol: Survey on attitudes and experiences regarding preregistration in psychological research 

Dear Dr. Spitzer:

I'm pleased to inform you that your manuscript has been deemed suitable for publication in PLOS ONE. Congratulations! Your manuscript is now with our production department. 

Kind regards, 

on behalf of

Pr. Florian Naudet 

Academic Editor

PLOS ONE